# Altered corticolimbic connectivity reveals sex-specific adolescent outcomes in a rat model of early life adversity

Jennifer A Honeycutt[1]*, Camila Demaestri[1], Shayna Peterzell[1], Marisa M Silveri[2,3], Xuezhu Cai[4], Praveen Kulkarni[4], Miles G Cunningham[5], Craig F Ferris[4], Heather C Brenhouse[1]*

[1]Developmental Neuropsychobiology Laboratory, Department of Psychology, Northeastern University, Boston, United States; [2]Neurodevelopmental Laboratory on Addictions and Mental Health, McLean Hospital, Belmont, United States; [3]Department of Psychiatry, Harvard Medical School, Boston, United States; [4]Center for Translational Neuroimaging, Department of Psychology, Northeastern University, Boston, United States; [5]Laboratory for Neural Reconstruction, Department of Psychiatry, McLean Hospital, Belmont, United States

**Abstract** Exposure to early-life adversity (ELA) increases the risk for psychopathologies associated with amygdala-prefrontal cortex (PFC) circuits. While sex differences in vulnerability have been identified with a clear need for individualized intervention strategies, the neurobiological substrates of ELA-attributable differences remain unknown due to a paucity of translational investigations taking both development and sex into account. Male and female rats exposed to maternal separation ELA were analyzed with anterograde tracing from basolateral amygdala (BLA) to PFC to identify sex-specific innervation trajectories through juvenility (PD28) and adolescence (PD38;PD48). Resting-state functional connectivity (rsFC) was assessed longitudinally (PD28;PD48) in a separate cohort. All measures were related to anxiety-like behavior. ELA-exposed rats showed precocial maturation of BLA-PFC innervation, with females affected earlier than males. ELA also disrupted maturation of female rsFC, with enduring relationships between rsFC and anxiety-like behavior. This study is the first providing both anatomical and functional evidence for sex- and experience-dependent corticolimbic development.

**\*For correspondence:**
j.honeycutt@northeastern.edu (JAH);
h.brenhouse@neu.edu (HCB)

## Introduction

Exposure to early life adversity (ELA) increases vulnerability to various psychiatric disorders across the lifespan (*Smyke et al., 2007*; *McEwen, 2008*; *Maccari et al., 2014*; *Callaghan and Tottenham, 2016*; *Hane and Fox, 2016*; *Krugers et al., 2017*). Males and females appear to be affected differently by ELA, with females more prone to developing disorders including anxiety and depression (*Hammen et al., 2000*; *Heim et al., 2008*; *Davis and Pfaff, 2014*). Importantly, these sequelae often emerge later in childhood or adolescence, providing an opportunistic window for intervention before psychopathology takes hold. Thus, development of effective intervention strategies requires biological and developmental targets specific to individuals based on factors including sex and timing of stress exposure (*Lupien et al., 2009*). Evidence suggests that ELA in humans leads to life-long changes in connectivity and/or functionality of limbic and cortical regions (*Choi et al., 2009*; *VanTieghem and Tottenham, 2018*), with consequential deficits in emotion regulation and cognition (*Tyrka et al., 2013*). Both human and animal studies highlight the importance of corticolimbic circuitry in affective behavior regulation, its disruption in mental disorders (*Herringa et al., 2013*; *Bangasser and Valentino, 2014*; *Killgore et al., 2014*), and alterations directly related to ELA

**eLife digest** Having a traumatic childhood increases the risk a person will develop anxiety disorders later in life. Early life adversity affects men and women differently, but scientists do not yet know why. Learning more could help scientists develop better ways to prevent or treat anxiety disorders in men and women who experienced childhood trauma.

Anxiety occurs when threat-detecting brain circuits turn on. These circuits begin working in infancy, and during childhood and adolescence, experiences shape the brain to hone the body's responses to perceived threats. Two areas of the brain that are important hubs for anxiety-related brain circuits include the basolateral amygdala (BLA) and the prefrontal cortex (PFC).

Now, Honeycutt et al. show that rats that experience early life adversity develop stronger connections between the BLA and PFC, and these changes occur earlier in female rats. In the experiments, one group of rats was repeatedly separated from their mothers and littermates (an early life trauma), while a second group was not. Honeycutt et al. examined the connections between the BLA and PFC in the two groups at three different time periods during their development: the juvenile stage, early adolescence, and late adolescence.

The experiments showed stronger connections between the BLA and PFC begin to appear earlier in juvenile traumatized female rats. But these changes did not appear in their male counterparts until adolescence. Lastly, the rats that developed these strengthened BLA-PFC connections also behaved more anxiously later in life. This may mean that the ideal timing for interventions may be different for males and females. More work is needed to see if these results translate to humans and then to find the best times and methods to help people who experienced childhood trauma.

(*Kaiser et al., 2018*). Specifically, Tottenham and colleagues illustrated that normative developmental changes in task-based functional connectivity (FC) between amygdala and medial prefrontal cortex (mPFC) are accelerated after ELA with associated changes in anxiety (*Gee et al., 2013a*) and are particularly evident in females (*Dickie and Armony, 2008*). Indeed, children institutionalized during the first two years of life in orphanages display *precocial* development of amygdala-mPFC FC, though the anatomical substrates for this accelerated connectivity remain unknown.

In rats, the basolateral amygdala (BLA) sends inputs to the mPFC and modulates anxiety-related behaviors (*Felix-Ortiz et al., 2016*), recall of emotionally salient information (*McGaugh, 2004*), decision-making (*St Onge et al., 2012*), and goal-directed behavior (*Schoenbaum et al., 2000*). Notably, two subpopulations of BLA neurons project to different regions of the mPFC (*Senn et al., 2014*): projections to the dorsal (prelimbic; PL) region of the mPFC are active during threat-associated fear learning and expression, whereas projections from the BLA to the ventral (infralimbic; IL) region of the mPFC are active upon extinction of fear, or learning about safety signals. In typically developing male rats, BLA innervation of mPFC increases through adolescence (*Cunningham et al., 2002*), likely contributing to healthy maturation of threat and safety appraisal. Notably, BLA-mPFC axonal innervation has not yet been evaluated in females, nor has any study examined the effects of ELA on BLA-derived innervation of PL or IL.

Evidence in both the human and rodent literature supports the role of amygdala-PFC circuitry in the regulation of anxiety-like behaviors (*Likhtik et al., 2014*; *Arruda-Carvalho and Clem, 2015*; *Fragale et al., 2016*). However, the neurobiological mechanisms are ill-defined and therefore require investment in translational animal research to pinpoint neural contributions to affective pathology (*Shackman and Wager, 2019*). Emerging evidence in rodents suggests that perturbations of this circuit – particularly following various models of ELA – may be implicated in maladaptive maintenance of anxiety-like behaviors (*Chocyk et al., 2013*; *Maccari et al., 2014*; *Krugers et al., 2017*). ELA via chronic stress has been shown to increase BLA-derived glutamatergic release within the PFC, a finding that can be recapitulated via BLA-PFC stimulation in typical mice and is associated with increased anxiety-like behavior (*Lowery-Gionta et al., 2018*). This is in line with reports of altered excitatory:inhibitory balance within the BLA-PFC circuit that is associated with changes in affective behaviors (*Arruda-Carvalho and Clem, 2014*). For example, excitatory latencies of PFC neurons to amygdalar stimulation in adult rats were significantly longer in animals with a history of ELA (*Ishikawa et al., 2015*). It is likely that ELA-induced changes in plasticity lead to increased BLA-

derived excitatory signaling into the PFC, with insufficient reciprocal PFC-driven anxiolytic signals returning to the BLA. However, there is presently insufficient evidence to determine the neuroanatomical time-course of this developing circuit, and a lack of evidence evaluating its modulation by additional factors (i.e. sex, development).

Clinical evidence points to striking sex differences in the clinical time-course and symptomology of ELA effects (*Wainwright and Surtees, 2002*; *Martin et al., 2014*). Moreover, several studies in animals over the past two decades have revealed sex-specific effects of ELA on anxiety-like behaviors (*Tractenberg et al., 2016*; *Bonapersona et al., 2019b*) and adolescent or adult corticolimbic measures (*Salzberg et al., 2007*; *Holland et al., 2014*; *Farrell et al., 2016*; *Blaze and Roth, 2017*; *Bonapersona et al., 2019a*), however little is known about the interaction of sex and ELA on corticolimbic development. Notably, typically developing females display earlier maturation of the PFC (*Lenroot et al., 2007*; *Lenroot and Giedd, 2010*). Identifying how ELA affects these sex-dependent trajectories is crucial to understanding sex differences in vulnerability and to developing individually targeted intervention strategies. Therefore, we used anterograde tracing to examine ELA effects on BLA-PFC innervation over development in male and female rats. We hypothesized that if heightened anxiety-like behaviors following ELA are associated with increased BLA-derived PFC innervation, then rats exposed to ELA via repeated isolation from dam and littermates will display anxiety-like behavior and increased BLA-PFC innervation that will be more robust in females.

While task-based FC illustrates coordinated responsivity to anxiety-provoking stimuli (*Gee et al., 2013a*), resting-state FC (rsFC) is excellent for probing the functional integrity of the amygdala-PFC circuit independent of task demands (*Thomason et al., 2011a*; *Thomason et al., 2011b*; *Alarcón et al., 2015*; *Gabard-Durnam et al., 2016*). ELA effects on corticolimbic rsFC in humans are inconsistent, likely because of reliance on autobiographical questionnaires, different ages of measurement, and different ELA criteria. In rats, maternal separation results in early emergence of both adult-like fear learning based in fronto-amygdala circuitry (*Callaghan and Richardson, 2011*) and early amygdala structural maturation (*Ono et al., 2008*). rsFC has recently been utilized in rats to characterize ELA (via limited bedding) induced changes in BLA-PFC connectivity in preweanling males, revealing an association between decreased rsFC and adult fear behaviors (*Guadagno et al., 2018*). Early or blunted maturation of corticolimbic connectivity likely has deleterious consequences because a sufficient degree of PFC immaturity during juvenility is critical for learning anxiolytic cues (safety signals) in adulthood (*Yang et al., 2012*). Therefore, we provide a back-translation to examine whether ELA-exposed rats display accelerated maturation of amygdala-PFC connectivity to parallel humans with a history of adversity, with increased BLA-PFC innervation as a potential anatomical substrate driving sex-specific developmental effects. This multi-level approach aims to elucidate the neurobiological underpinnings of ELA-attributable vulnerability through the use of mechanistic neural tracing paired with a translational imaging investigation: an approach which enhances our ability to develop a cross-species understanding of ELA-associated pathology (*Fox and Shackman, 2019*).

## Results

### Study 1: Neuroanatomy and axonal innervation

In order to chart the trajectory of innervation from the BLA to the PFC, the anterograde tracer biotinylated dextran amine (BDA) was injected into the BLA of ELA-exposed and control (CON) males and females at PD21, PD31, or PD41. Seven days following surgery (PD28, PD38, or PD48), animals were assessed for anxiety-like behavior in the elevated plus maze (EPM) and then sacrificed for quantification of BLA axonal terminals in the PFC.

### BLA volume

A subset of tissue from each group was analyzed for BLA volume with Cresyl Violet. Average BLA volume for each age (PD28, PD38, PD48) was calculated using the Cavalieri probe in StereoInvestigator ($n$ = 16 per age). Two-way ANOVA for each age revealed no effects of sex or rearing on BLA volume (*Figure 1—figure supplement 1*). Thus, results were pooled across sex and rearing to determine BLA volume across development. One-way ANOVA showed a main effect of age ($F_{2,48}$ = 31.74, p<0.0001). Comparisons revealed increased BLA volume between PD28 and PD38 (p=0.002),

PD28 and PD48 (p<0.0001); and PD38 and PD48 (p=0.0002) (*Figure 1A*; BLA anatomical examples can be seen in *Figure 1B*).

## BLA bolus

Standard deviation of BLA volumes within each age deviated ~ 1% from mean volume, therefore percent BDA-filled BLA was calculated by dividing bolus volume in BLA in each animal by average BLA volume per age. Percent bolus outside BLA was Since age differences were detected in BLA volume, we examined whether amount of BLA filled differed with age. One-way ANOVA revealed no effect of percent BLA filled ($F_{2,88}$ = 3.030, p=0.053; small effect size: partial $\eta^2$ = 0.06) (*Figure 1C*), and no effect of percent bolus outside of BLA across age ($F_{2,88}$ = 0.133, p=0.876) (*Figure 1D,E*). Source data is provided in *Figure 1—source data 1*.

To ensure that percentage of bolus within BLA did not drive individual differences in PFC axonal innervation, linear regressions were used to determine correlations between percent of filled BLA volume and total innervation in the PFC, as well as the percent of BLA filled with tracer and total innervation in the PFC. These revealed no relationship between percent BLA filled and total innervation at any age (PD28: $R^2$(28)=0.064, p=0.178; PD38: $R^2$(27)=0.004, p=0.738; PD48: $R^2$(30)=0.057, p=0.189). Additionally, linear regression analyses were used to investigate possible relationships between the total bolus volume (both within and outside of the BLA) and total PFC innervation,

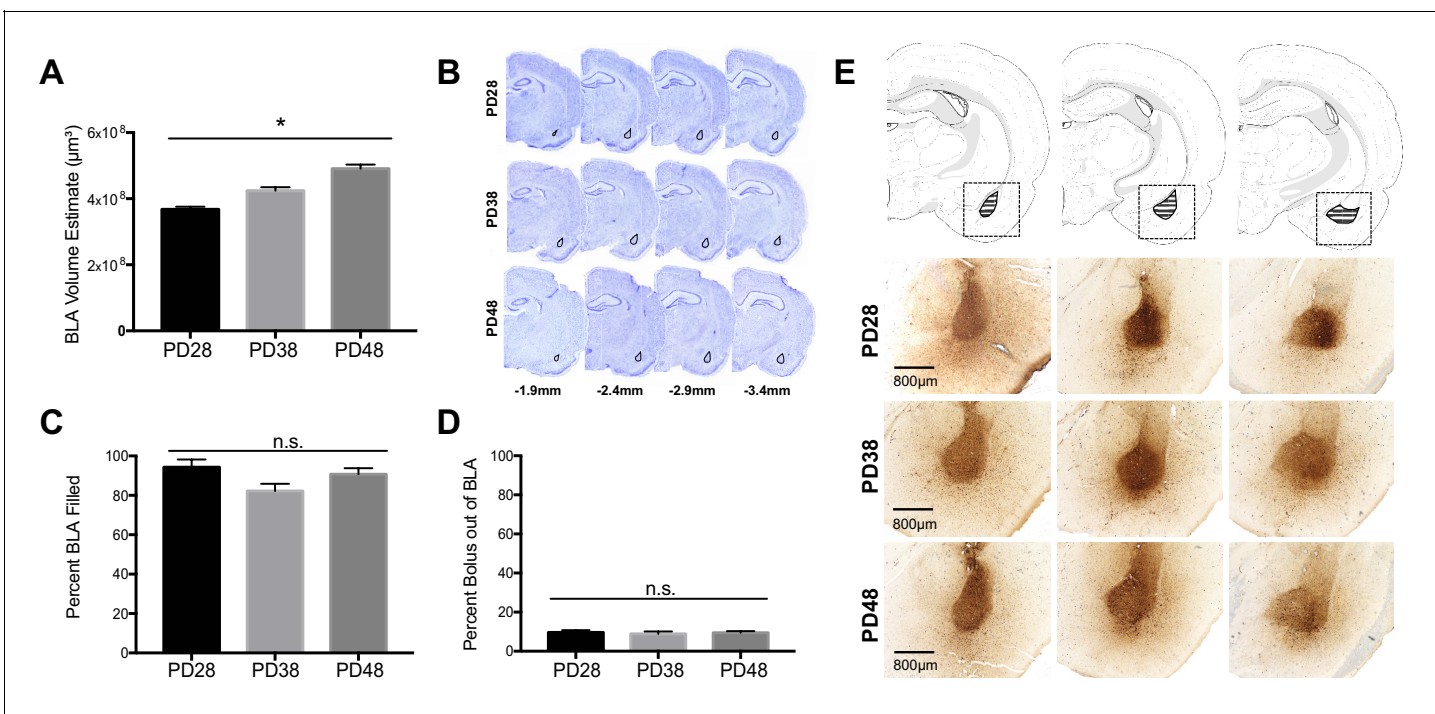

**Figure 1.** BLA volume and BDA anterograde tracer bolus sites. The average volume of the BLA significantly increased with age (**A**) as quantified via Cresyl Violet staining of adjacent tissue sections (**B**). As there were no significant differences as a function of sex or rearing condition on BLA volume estimates at each age group, data were collapsed for each age with a final number of n = 16 per age group. Anatomical coordinates located below Nissl-stained sections (**B**) indicate the approximate distance (in mm) from Bregma. There were no significant differences in the percent of BLA filled by the BDA anterograde tracer bolus between ages (**C**), nor was there any significant difference in the amount of bolus located outside of the BLA structure (**D**). n = 6–9 per group (**C**; **D**) before collapsing within each age due to a lack of differences as a function of sex or rearing condition at each age (therefore, the number of data points per age group for analyses in **C** and **D** were 24–36. Panel (**E**) shows representative photomicrographs of BDA bolus across the extent of the BLA for each age examined. Atlas modified from *Swanson (2018)*.

The online version of this article includes the following source data and figure supplement(s) for figure 1:

**Source data 1.** Raw Bolus Volume Data.

**Figure supplement 1.** No effects of sex or rearing condition on BLA volume.

**Figure supplement 2.** PFC innervation is not driven by percentage of BLA filled or bolus size in included cases.

which revealed no significant correlation at any age (PD28: $R^2(28)$=0.062, p=0.185; PD38: $R^2(27)$ =0.006, p=0.696; PD48: $R^2(30)$=0.090, p=0.096) (*Figure 1—figure supplement 2*).

## BLA-Derived axonal innervation

There were no differences in probe volume or mounted thickness (p>0.1) between groups based on age, sex, or rearing; thus, all analyses are presented as collected without corrections (*Figure 2—figure supplement 1*).

## PL innervation

A three-way ANOVA to determine effects of rearing, sex, and age on BLA-PL innervation revealed a main effect of rearing ($F_{1,79}$ = 14.294, p<0.0001; partial $\eta^2$ = 0.153), and a three-way interaction ($F_{2,79}$ = 5.244, p=0.007; partial $\eta^2$ = 0.116). No main effect of age (p=0.763; partial $\eta^2$ = 0.006) or sex (p=0.660; partial $\eta^2$ = 0.002) was found. Two-way ANOVA of male PL innervation showed a trending age x rearing interaction ($F_{2,42}$ = 2.748, p=0.076) with a moderate effect size (partial $\eta^2$ = 0.116); follow-up post-hoc showed an ELA-driven increase in innervation at PD38 compared to CON (p=0.031). Increased innervation at PD38 compared to PD28 was observed in ELA (p=0.044). No main effect of age (p=0.260; partial $\eta^2$ = 0.064) or rearing (p=0.171; partial $\eta^2$ = 0.045) was observed in male PL innervation. Two-way ANOVA of female PL innervation showed a main effect of rearing ($F_{1,38}$ = 14.21, p<0.001; partial $\eta^2$ = 0.272), with post-hoc indicating more PL innervation in ELA compared to CON at PD28 (p=0.009) and PD48 (p=0.010). Age x rearing interaction in female PL innervation was not statistically significant but revealed a moderate effect size (p=0.091; partial $\eta^2$ = 0.119). No main effect of age was observed (p=0.469; partial $\eta^2$ = 0.039). Graphs detailing comparisons, as well as representative photomicrographs of PL, can be seen in in *Figure 2A*.

Three-way ANOVAs to delineate layer-specific effects in PL revealed a main effect of rearing in PL2 ($F_{1,79}$ = 12.511, p=0.001 partial $\eta^2$ = 0.137) and PL5 ($F_{1,79}$ = 12.925, p=0.001; partial $\eta^2$ = 0.141). There were significant rearing x sex x age interactions in PL2 ($F_{2,79}$ = 4.655, p=0.012; partial $\eta^2$ = 0.105) and PL5 ($F_{2,79}$ = 4.699, p=0.012; partial $\eta^2$ = 0.106). In PL5, a rearing x sex interaction was observed ($F_{1,79}$ = 7.775, p=0.007; partial $\eta^2$ = 0.090). Two-way ANOVAs were conducted for each sex to determine the impact of rearing condition and age. We observed no main effects or interactions of age or rearing within PL2 or PL5 in males that met the Bonferroni-corrected $\alpha$ of 0.025 (see *Supplementary file 1*). However, two-way ANOVA in females revealed a main effect of rearing ($F_{1,38}$ = 8.11, p=0.007; partial $\eta^2$ = 0.176) in PL2 and a main effect of rearing ($F_{1,38}$ = 17.760, p<0.0001; partial $\eta^2$ = 0.319) and a trend-level age x rearing interaction ($F_{2,38}$ = 2.987, p=0.062; partial $\eta^2$ = 0.136) in PL5. Post-hoc comparisons revealed an effect of rearing with female ELA showing more PL5 innervation than CON at PD28 (p=0.002) and PD48 (p=0.007). No main effect of age on female PL2 innervation (p=0.576; partial $\eta^2$ = 0.029) or PL5 innervation (p=0.327; partial $\eta^2$ = 0.057) was found.

## IL innervation

Three-way ANOVA revealed no overall main effects or interactions in the IL as a whole (see *Supplementary file 1*). However, since our a priori hypothesis was that the effects of ELA would be sex specific, we also performed separate 2-way ANOVAs for males and females. Two-way ANOVA in males showed a main effect of age ($F_{2,40}$ = 4.355, p=0.019; partial $\eta^2$ = 0.132), with increased IL innervation from PD28 to PD38 (p=0.015) in ELA. No main effect of rearing (p=0.547; partial $\eta^2$ = 0.009) or age x rearing interaction (p=0.410; partial $\eta^2$ = 0.043) was found in males. Two-way ANOVA of IL innervation in females showed no main effects or an age x rearing interaction (see *Supplementary file 1*). Graphs detailing comparisons, as well as representative photomicrographs of IL, can be seen in in *Figure 2B*.

Three-way ANOVAs were conducted for IL2 and IL5. Adjusted $\alpha$ was set to 0.025 to correct for multiple comparisons. Of note, two data points from male IL2 counts and one data point from male IL5 counts were eliminated from analyses due to lack of confidence in the ROI's chosen. In IL2, a main effect of rearing was evident ($F_{1,79}$ = 5.938, p=0.017; partial $\eta^2$ = 0.070), with no main effect of age (p=0.122; partial $\eta^2$ = 0.052) or 3-way interaction (p=0.155; partial $\eta^2$ = 0.046). No main effects or 3-way interactions were observed in IL5 (see *Supplementary file 1*). Two-way ANOVAs were conducted for each sex and sub-region. In IL2, Males displayed a main effect of age ($F_{2,39}$ = 4.411,

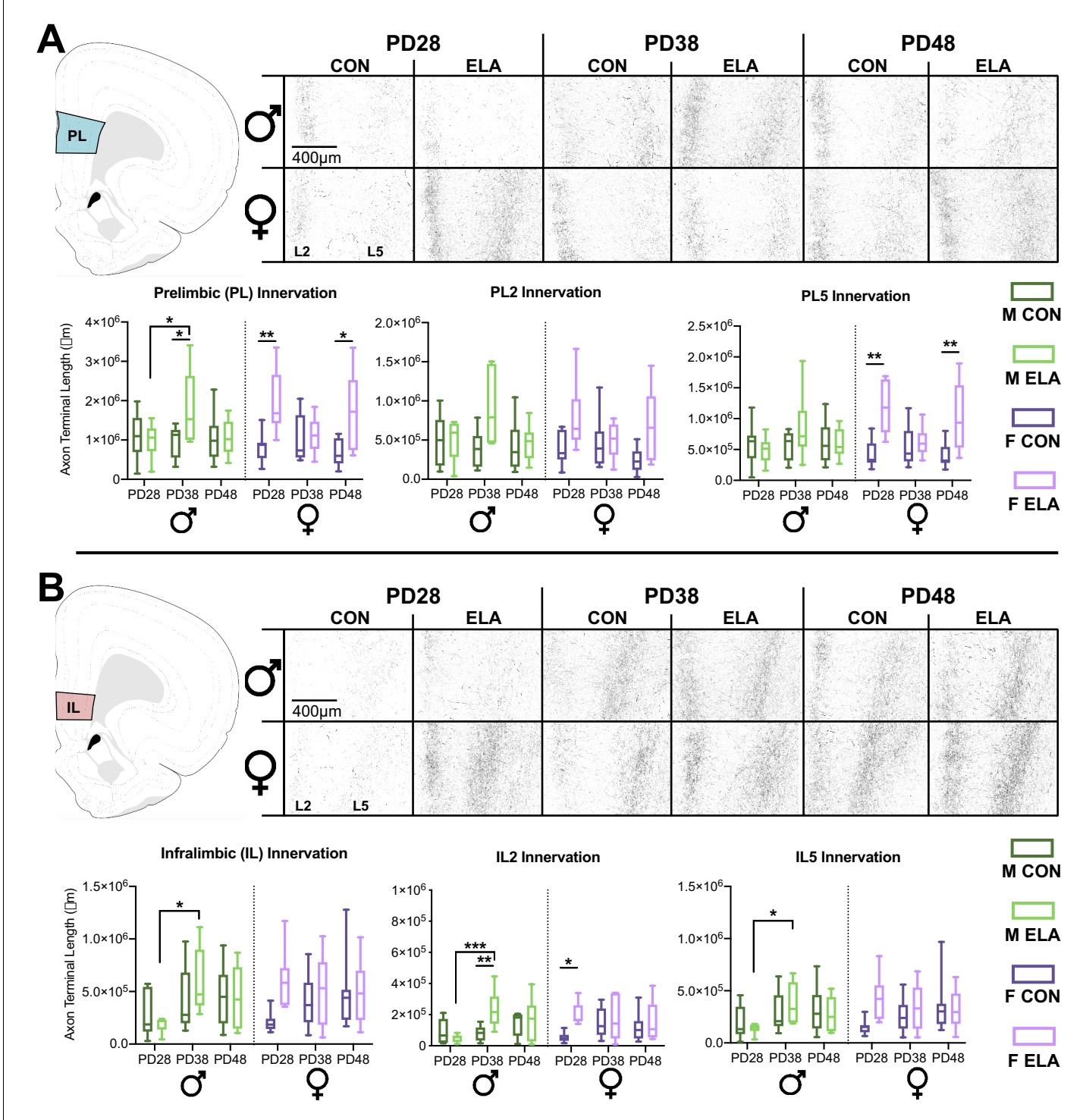

**Figure 2.** ELA leads to precocial BLA-PFC axonal innervation earlier in female than in male rats. BLA-PFC axonal innervation of anterogradely biotinylated dextran amine (BDA)-labeled fibers were visualized with diaminobenzidine (DAB) staining, quantified via unbiased stereology, and are displayed here as an estimate of total axon terminal length (μm) in male and female rats in both PL (**A**) and IL (**B**). Top panel (**A**) indicates PL quantification location, as well as the representative photomicrographs, with dark axonal fibers clearly observed in layer 2 (L2; left side of each photomicrograph) and layer 5 (L5; right side of each photomicrograph). In the PL there was consistent innervation across CON groups at all ages, with transient ELA-induced spikes of innervation occurring at PD28 and PD48 in female rats, and at PD38 in male rats. This pattern of innervation appears to be driven by PL5 axon terminal length, particularly in ELA females. Bottom panel (**B**) indicates IL quantification location, as well as representative photomicrographs similar to those seen in PL. In the IL, female rats showed a precocial pattern of axonal innervation by PD28 that was comparable to

*Figure 2 continued on next page*

Figure 2 continued

adolescents and older adolescents, while male rats didn't show an ELA-driven increase in IL innervation until PD38, with these findings appearing to be driven by IL2 axon terminal length, particularly in ELA animals. Data is presented as a function of sex (male, female), age at brain collection (PD28, PD38, PD48), and rearing condition (CON, ELA), with lines inside each group bar indicating group mean, and lines outside of the bars indicating the maximum and minimum observed data points within that group. $n$ = 6–9 per group. The left graph for both PL (**A**) and IL (**B**) displays collapsed L2 and L5 data for the entire quantified region, with the middle and right graphs displaying layer-specific data (alpha adjusted to 0.025 significance threshold to account for multiple comparisons in subsequent two-way ANOVAs). Photomicrographs were imaged at 10x magnification. Atlas modified from *Swanson (2018)*. *$p<0.05$ (for IL/PL Innervation graphs); *$p<0.025$ (for individual L2 and L5 innervation graphs); **$p<0.01$; ***$p<0.001$.

The online version of this article includes the following source data and figure supplement(s) for figure 2:

**Source data 1.** Raw Histological and Probe Data.
**Figure supplement 1.** No differences in mounted thickness or probe volume.

$p=0.019$; partial $\eta^2$ = 0.184), and a rearing x age interaction ($F_{2,39}$ = 4.521, $p=0.017$; partial $\eta^2$ = 0.188). Male ELA showed more IL2 innervation compared to CON at PD38 ($p=0.007$). Developmentally, male ELA had more innervation at PD38 than PD28 ($p<0.001$). In females, there was a trending main effect of rearing ($F_{1,38}$ = 4.791, $p=0.035$) with a moderate effect size (partial $\eta^2$ = 0.126), with more IL2 innervation in ELA than CON at PD28 ($p=0.037$). No rearing x age interaction was noted for female IL2 innervation ($p=0.247$; partial $\eta^2$ = 0.071). Two-way ANOVA on male IL5 innervation showed a trending main effect of age ($F_{2,40}$ = 3.541, $p=0.038$; partial $\eta^2$ = 0.178), with post-hoc comparisons revealing more innervation at PD38 than PD28 in ELA ($p=0.048$). Two-way ANOVA on female IL5 showed no main effects or interaction (see supplemental table for *Figure 2*). Source data is provided in *Figure 2—source data 1*.

## EPM

Six-seven days following surgery, subjects were analyzed for time spent in the open arms of an elevated plus maze (EPM) as a measure of anxiety-like behavior. Three-way ANOVA comparing time spent in the open arms showed a small effect for a rearing x sex interaction ($F_{2,238}$ = 4.989; $p=0.026$; partial $\eta^2$ = 0.02) as well as a main effect of rearing ($F_{2,238}$ = 15.833; $p=0.004$; partial $\eta$ = 0.04) and of sex ($F_{2,238}$ = 7.494; $p=0.007$; partial $\eta^2$=0.03), with no 3-way interaction ($p=0.532$; partial $\eta^2$=0.001) (*Figure 3A*). Two-way ANOVAs revealed a main effect in females of rearing ($F_{1,132}$ = 15.833, $p<0.0001$; partial $\eta^2$=0.107), with ELA females spending less time in the open arms than CON at PD38 ($p=0.0176$). No effect of age ($p=0.253$; partial $\eta^2$ = 0.021) or age x rearing interaction ($p=0.409$; partial $\eta^2$ = 0.013) was observed for open arm time in females. No effects of treatment or age were observed in males (see *Supplementary file 2*). Arm crossings and head dips were also analyzed. No interactions or main effects of age, sex, or rearing were found with a 3-way ANOVA on head dips (*Figure 3B*), however an age x sex interaction ($F_{2,242}$ = 6.73; $p=0.001$) for arm crosses and follow-up 2-way ANOVA in males revealed a main effect of age in males (main effect of age: $F_{2,114}$ = 7.22; $p=0.007$), with fewer arm crosses at PD48 than at PD28 ($p=0.0006$) or PD38 ($p=0.0203$) (*Figure 3C*). Source data is provided in *Figure 3—source data 1*.

## Relationships between innervation and behavior

Results from all analysis of correlation between behavior on the EPM and connectivity measures are shown in *Table 1*. Here we discuss significant correlations, which are illustrated in *Figure 4A–D*. Fisher's r- to- z transformations revealed an impact of sex, but not rearing, on the strength of some relationships between innervation and behavior (*Table 1*). At PD28, higher IL innervation in females was correlated with less time spent in the open arms ($R^2(14)=0.478$; $p=0.009$) (*Figure 4B*). At PD38, males showed a similar relationship between less time in the open arms and higher PL innervation ($R^2(13)=0.266$; $p=0.049$) (*Figure 4C*) and higher IL innervation ($R^2(13)=0.260$; $p=0.05$) (*Figure 4D*). No relationships were seen at PD48 (*Table 1*).

## Study 2 results: rsFC

A separate cohort of CON and ELA-exposed males and female rats were subjected to anxiety-like behavioral assessment in the EPM, as well as MRI scanning, at both PD28 and PD38 using a

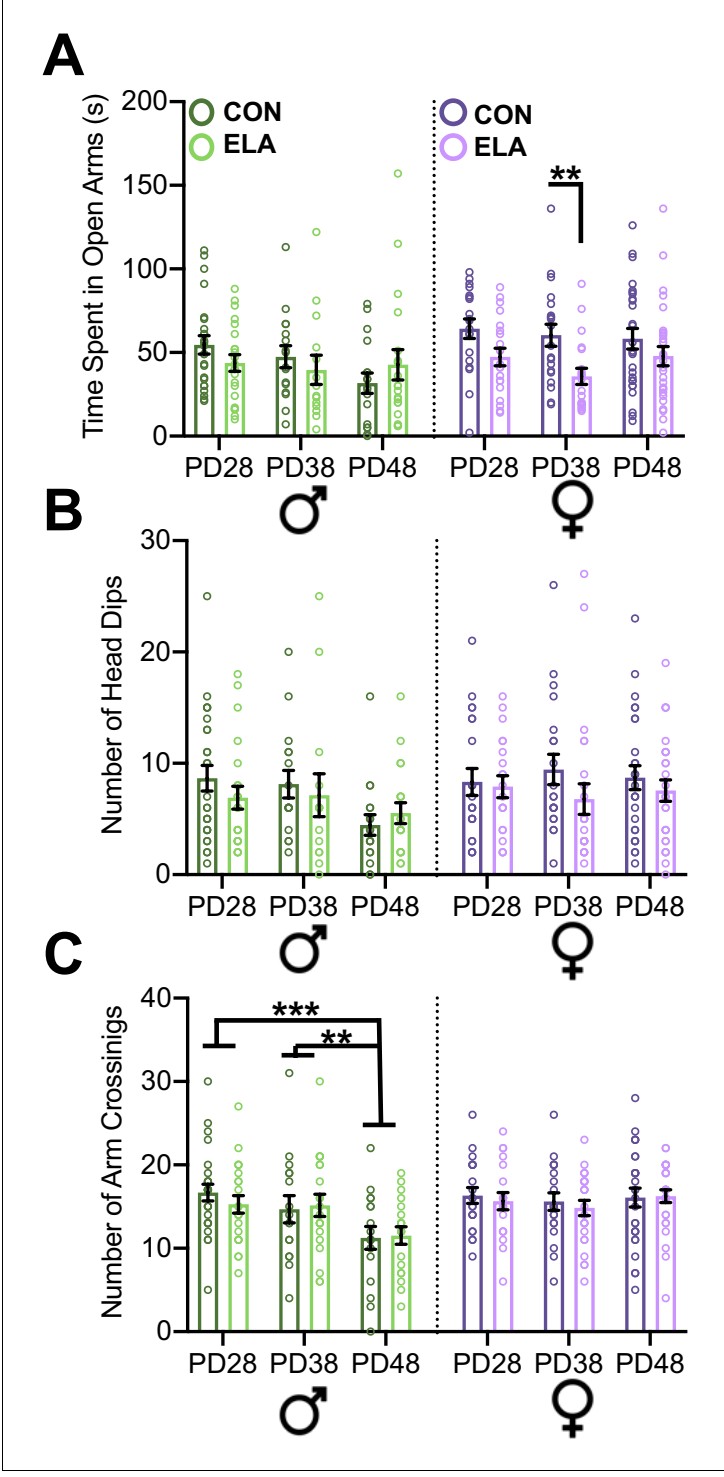

**Figure 3.** Anxiety-like behavior following ELA is sex- and age-specific. Anxiety-like behavior was assessed in the elevated plus maze (EPM) in both CON male (dark green bard) and female (dark purple bars) and ELA male (light green bars) and female (light purple bars) groups. Anxiety-like behavior in the EPM was assessed via time spent in open arms (seconds; A), number of head dips (B), and number of arm crossings (C). ELA animals overall displayed more anxiety-like behavior (measured via less time spent in open arms). Female, but not male, rats showed more anxiety-like behavior, exclusively at PD38 if they had been exposed to ELA (A). Analysis of the number of arm crossings (C) revealed that only males showed a significant effect of age, with PD48 males showing less arm crossings (indicative of increased anxiety) than those at PD28 and PD38. Each circle indicates the data from a
*Figure 3 continued on next page*

*Figure 3 continued*

single animal, and data shown is for behavior collected from all subjects, including those that did not meet neuroanatomical inclusion criteria. *n* = 16-28 per group.*p < 0.05; **p < 0.01; ***p < 0.001 ".

The online version of this article includes the following source data for figure 3:

**Source data 1.** Raw Study 1 Elevated Plus Maze Data.

longitudinal design. rsFC was analyzed with the BLA as a seed, and correlation coefficients between the BLA and either PL or IL were compared between rearing groups and age within each sex.

## Males

Two-way ANOVA evaluating BLA-PL rsFC (analyses of these regions were seeded to the BLA; *Figure 5A–B*) as a function of age and rearing condition revealed no main effect of ELA, but a significant main effect of age ($F$ = 49.9), in addition to two regions with an age x rearing interaction in PL that met threshold criteria ($F$ = 15.8; $F$ = 28.6). Post-hoc comparisons showed reduced strength of BLA-PL rsFC from PD28 to PD48 in both CON ($Z$ = −3.82) and ELA ($Z$ = −3.26) rats. There was no difference in rsFC between CON and ELA at PD48; however, at PD28 CON showed stronger BLA-PL connectivity than ELA ($Z$ = −4.13; *Figure 5C*). A similar 2-way ANOVA evaluating BLA-IL rsFC revealed a significant main effect of age ($F$ = 49.9), with no post-hoc differences. Effect size maps from CON-ELA comparisons are illustrated in *Figure 5—figure supplement 1* and full maps can be downloaded from Dryad: https://doi.org/10.5061/dryad.jdfn2z371.

## Females

Two-way ANOVA (age x rearing) for BLA-PL revealed a main effect of age ($F$ = 13.76) and rearing ($F$ = 16.10), and an interaction ($F$ = 12.65). Overall, BLA-PL rsFC decreased from PD28 to PD48 ($Z$ = −3.6), while ELA females displayed stronger connectivity compared to CON ($Z$ = 3.5). Post-hoc showed no difference between CON and ELA at PD28, though a significant difference at PD48 when ELA had stronger BLA-PL connectivity than CON ($Z$ = 4.83; *Figure 5C*). CON connectivity decreased from PD28 to PD48 ($Z$ = −3.09). Conversely, ELA BLA-PL connectivity increased from PD28 to PD48 ($Z$ = 3.46).

Two-way ANOVA for BLA-IL rsFC showed no main effect of age or rearing, though an interaction was observed ($F$ = 28.74). Overall, BLA-IL rsFC increased from PD28 to PD48 ($Z$ = 3.35), and CON

**Table 1.** Relationships between BLA-PFC innervation and anxiety-like behavior across development.

| | | PL Innervation vs. | | IL Innervation vs. | |
|---|---|---|---|---|---|
| | | Time in Open | | Time in Open | |
| PD28 | MALE | $R$ = 0.209; $R^2$ = 0.044 | | MALE | $R$ = 0.158; $R^2$ = 0.025 |
| | *n* = 16 | p=0.437 | | *n* = 14 | p=0.560 |
| | FEMALE | $R$ = 0.188; $R^2$ = 0.035 | | FEMALE | *R = 0.691; $R^2$ = 0.478* |
| | *n* = 14 | p=0.519 | | *n* = 12 | p=0.009** |
| PD38 | MALE | *R = 0.516; $R^2$ = 0.266* | | MALE | $R$ = 0.509; $R^2$ = 0.260 |
| | *n* = 15 | p=0.049* | | *n* = 13 | p=0.050* |
| | FEMALE | $R$ = 0.040; $R^2$ = 0.002 | | FEMALE | $R$ = 0.363; $R^2$ = 0.132 |
| | *n* = 14 | p=0.893 | | *n* = 12 | p=0.202 |
| PD48 | MALE | $R$ = 0.202; $R^2$ = 0.041 | | MALE | $R$ = 0.086; $R^2$ = 0.007 |
| | *n* = 16 | p=0.454 | | *n* = 14 | p=0.751 |
| | FEMALE | $R$ = 0.378; $R^2$ = 0.143 | | FEMALE | $R$ = 0.194; $R^2$ = 0.038 |
| | *n* = 16 | p=0.149 | | *n* = 14 | p=0.471 |

Bold: significant correlation without meeting criterion for significant sex effect.

Bold and Blue: significant correlation and significant sex difference (p<0.05).

Bold and Red: significant correlation and trend-level sex difference (p=0.058).

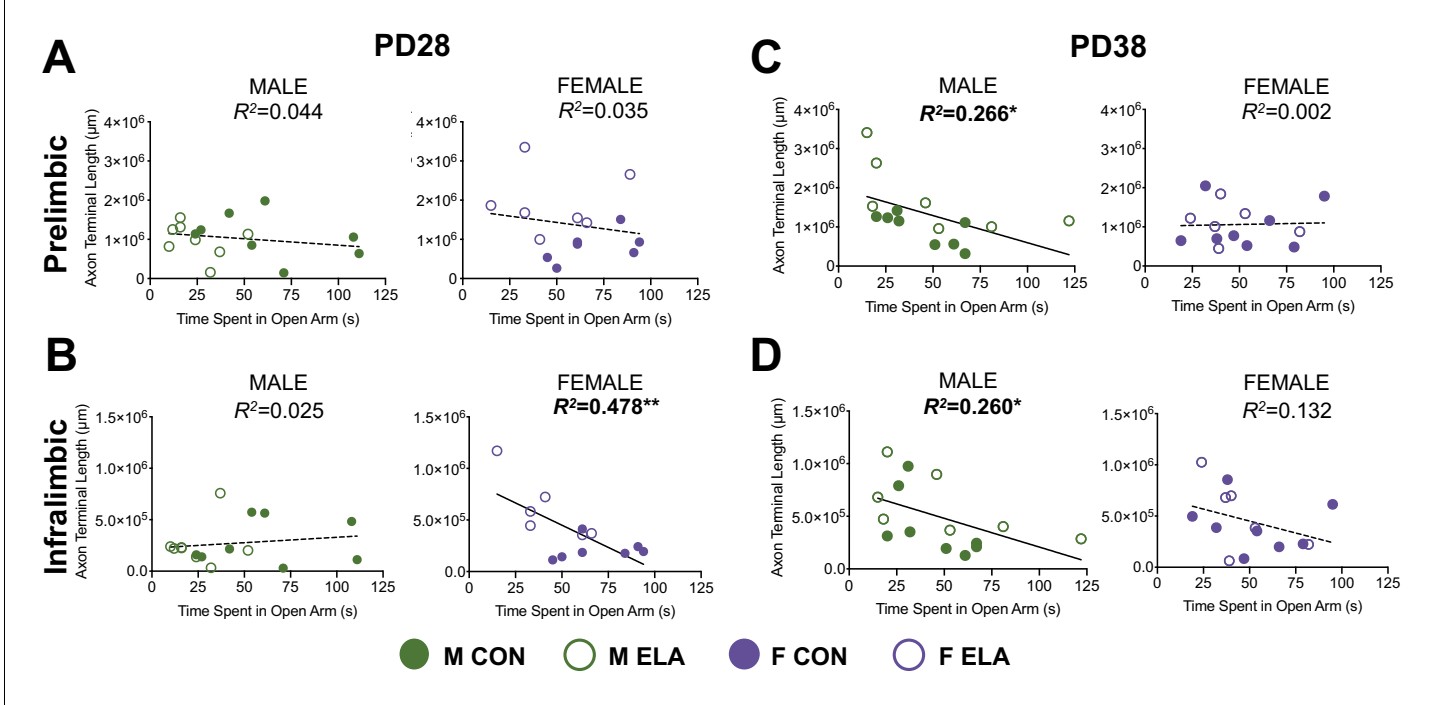

**Figure 4.** More BLA-derived axonal innervation into the PFC is correlated with increased anxiety-like behavior in a sex- and age-dependent manner. Results of linear regression analyses to determine the relationship between BLA-PFC innervation (measured as an estimate of BDA-labeled axon terminal length) and performance in the EPM (time spent in open arms (seconds) as an index of anxiety-like behavior). At PD28, there were no significant correlations between axon terminal length in the PL and time spent in open arms of the EPM (**A**) in either males or females, regardless of rearing condition. However, females, but not males, showed a significant correlation between IL innervation and time spent in open arms at PD28 (**B**), suggesting that more axonal innervation of the IL was related to increased anxiety-like behavior (via decreased time spent in open arms). Conversely, at PD38, females no longer show behavior correlated with innervation in either PL (**C**) or IL (**D**), whereas males generally show that increased axonal innervation is correlated with increased anxiety in both of these regions (**C**; **D**). Regression lines are reflective of the overall relationship across both groups (CON and ELA), and individual subjects from anatomically included cases are signified by either solid circles (CON) or open circles (ELA). Bold lines/regression statements indicate a significant correlation. $n$ = 6–9 per group. *p<0.05; **p<0.01.

generally had weaker connectivity compared to ELA (Z = −3.79). Post-hoc revealed that, at PD48, ELA showed less BLA-IL connectivity than CON (Z = −6.97; **Figure 5C**), and CON showed increased BLA-IL connectivity from PD28 to PD48 (Z = 4.47), with no significant change seen in ELA. Effect size maps from CON-ELA comparisons are illustrated in **Figure 5—figure supplement 1** and full maps can be downloaded from Dryad: https://doi.org/10.5061/dryad.jdfn2z371.

After noting what appeared to be a lack of typical BLA-PFC rsFC maturation in ELA-exposed females, we then analyzed the effects of sex and rearing on the magnitude of change in BLA-IL correlation coefficients and BLA-PL correlation coefficients between PD28 and PD48. Two-way ANOVA of BLA-IL revealed a moderate effect size for a trend-level sex x rearing interaction (partial $\eta^2$ = 0.167; $F_{1,25}$ = 4.09; p=0.054), and post-hoc revealed that females exposed to ELA displayed less PD28-PD48 change compared to CON females (p=0.033) (**Figure 5D**). In contrast, no effect of rearing was observed for the magnitude of change in BLA-PL rsFC (p=0.78) (**Figure 5E**).

## EPM

Prior to scanning on both PD28 and PD48, subjects were analyzed for time spent in the open arms of the EPM. Three-way mixed ANOVA with age as a repeated measure showed no effect of rearing at these ages ($F_{1,27}$ = 0.042; p=0.840; partial $\eta^2$ = 0.002), but, similarly to Study 1, revealed a main effect of sex ($F_{1,27}$ = 5.52; p=0.026; partial $\eta^2$ = 0.170; **Figure 6—figure supplement 1**). No rearing x sex interaction ($F_{1,27}$ = 0.096; p=0.754; partial $\eta^2$ = 0.003) or 3-way interaction ($F_{1,27}$ = 0.015; p=0.9; partial $\eta^2$ = 0.001) was observed. Source data is provided in **Figure 6—figure supplement 1—source data 1**.

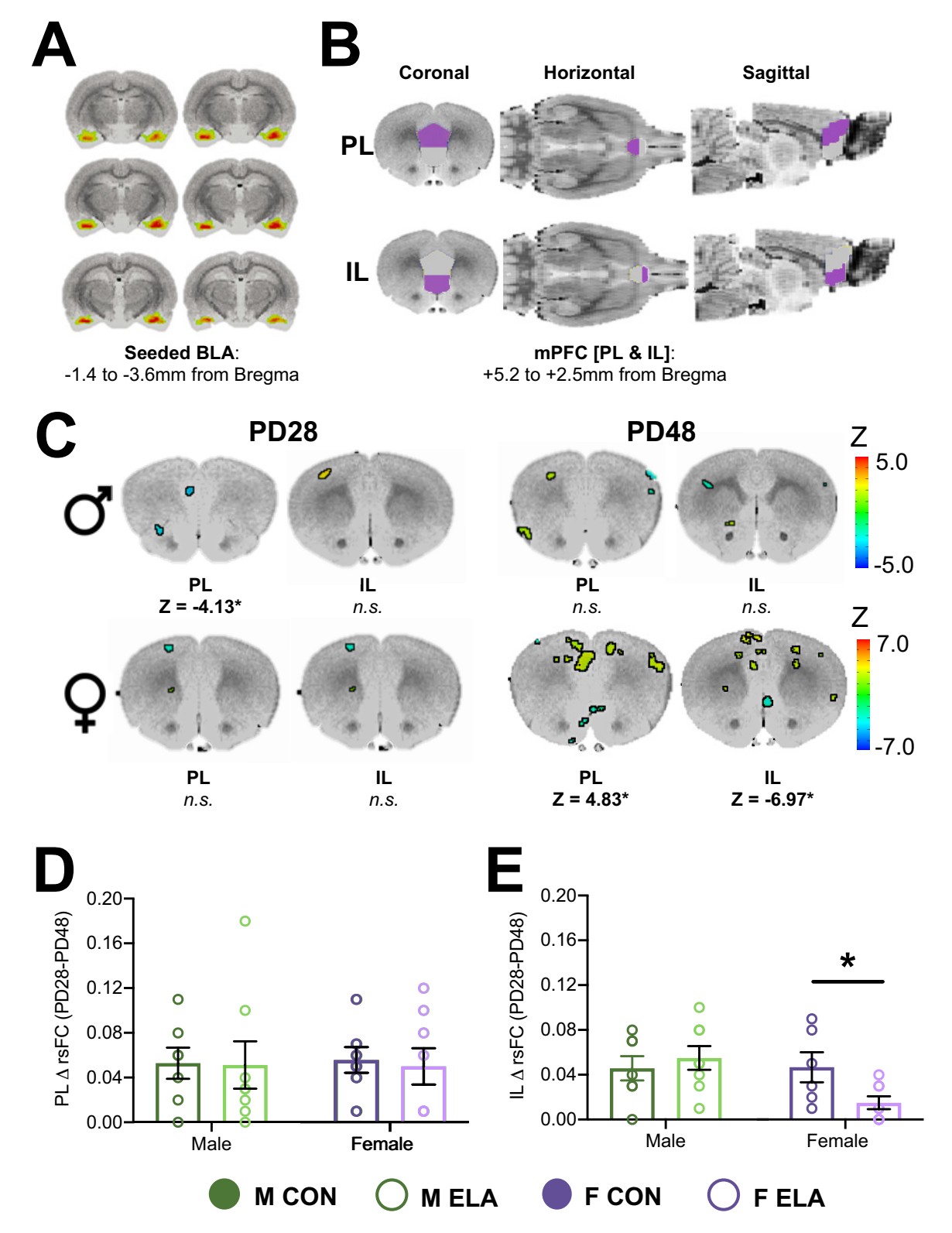

**Figure 5.** Effects of ELA on BLA-PFC rsFC are sex-specific and endure in females. Resting state functional connectivity (rsFC) was assessed in a longitudinal manner across development in male and female rats with a history of either CON or ELA rearing with the basolateral amygdala (BLA) as the seeded region (A). The rsFC between the BLA and the mPFC – specifically the prelimbic (PL) and infralimbic (IL) cortices (B) – was assessed. (C) Shows results from comparative analyses comparing CON and ELA groups at each age point (PD28 or PD48) in both male and female groups. Colored

*Figure 5 continued on next page*

Figure 5 continued

regions correspond to computed Z values and indicate specific regions within either the PL or IL that met criteria for significance with a minimum cluster size of 30 voxels. Generally, male rats show no group effects of ELA, with the exception of a finding of decreased BLA-PL rsFC in ELA compared to CON groups at PD28. Female rats showed no effect of ELA at PD28 but show striking differences in rsFC in both the PL and IL at PD48 (C). Effect size (Cohen's D) maps for corresponding sections are illustrated in *Figure 5—figure supplement 1*, and a full collection of maps are available on Dryad https://doi.org/10.5061/dryad.jdfn2z371. Female, but not male, rats exposed to ELA showed a lack of typical maturation of the IL, evidenced by significantly reduced ΔrsFC from PD28 to PD48 compared to CON females (D). Conversely, there were no group changes in ΔrsFC observed within the PL (E). *n* = 7–8 per group. *p<0.05.

The online version of this article includes the following figure supplement(s) for figure 5:

**Figure supplement 1.** Effect size maps for cortical sections comparing CON to ELS in males and females at PD28 or PD48.

## Relationships between rsFC and behavior

Results from all regression analyses of the relationship between behavior on the EPM and functional connectivity are shown in *Tables 2* and *3*. Fisher's r- to- z transformations revealed an impact of both sex and rearing on relationships between rsFC and behavior at PD28. Significant correlations revealed that in PD28 females exposed to ELA, lower BLA-IL rsFC predicted less time spent in the open arms ($R^2$(8)=0.852; p=0.001) (*Figure 6A*); notably, this relationship is juxtaposed with our finding that higher axonal innervation at the same age predicted less time spent in the open arms in all females (see *Figure 4B*). Our longitudinal design further allowed the analysis of early rsFC with later adolescent behavior. In all females, lower BLA-IL rsFC at PD28 also predicted less time spent in the open arms twenty days later at PD48 ($R^2$(7)=0.405; p=0.011), suggesting an enduring and predictive relationship in females (*Figure 6B*). This enduring relationship in females was also seen in a significant relationship between lower BLA-IL rsFC at PD48 and less time in the open arms at the same age (*Figure 6C*; $R^2$(13)=0.303; p=0.041).

## Discussion

This work identifies sex-specific neuroanatomical development and corresponding functional maturation of the BLA-PFC circuit following ELA. The data reveal newly uncovered aberrations to PFC innervation that can be interpreted in the context of atypical behavior and FC that has been observed in humans (*Gee et al., 2013a*; *Philip et al., 2013*; *Thomason et al., 2015*; *Teicher et al., 2016*) and in animals (*Yan et al., 2017*; *Johnson et al., 2018*). These findings are the first description of sex-specific developmental trajectories following ELA, using a juxtaposition of rsFC as assessed in humans with an anatomically discrete measurement of monosynaptic innervation. We observed sex- and age-dependent effects on BLA innervation in PL and IL regions of the PFC following ELA in a rat model of caregiver deprivation. It is important to note here, especially given the

**Table 2.** Relationships between BLA-PFC rsFC and anxiety-like behavior at each age.

| | | rsFC BLA-PL vs. Time in Open | | rsFC BLA-IL vs. Time in Open | |
|---|---|---|---|---|---|
| | | MALE | FEMALE | MALE | FEMALE |
| PD28 | CON | $R$ = 0.025; $R^2$ = 0.001 | $R$ = 0.212; $R^2$ = 0.045 | $R$ = 0.158; $R^2$ = 0.025 | $R$ = 0.184; $R^2$ = 0.034 |
| | $n$'s = 8 | p=0.953 | p=0.615 | p=0.709 | p=0.662 |
| | ELA | $R$ = 0.328; $R^2$ = 0.108 | $R$ = 0.390; $R^2$ = 0.152 | $R$ = 0.046; $R^2$ = 0.002 | $R$ = 0.923; $R^2$ = 0.852 |
| | $n$'s = 8 | p=0.427 | p=0.340 | p=0.913 | p=0.001* |
| PD48 | | $R$ = 0.026; $R^2$ = 0.001 | $R$ = 0.051; $R^2$ = 0.003 | $R$ = 0.001; $R^2$ < 0.001 | $R$ = 0.551; $R^2$ = 0.303 |
| | $n$'s = 16 | p=0.927 | p=0.863 | p=0.996 | p=0.041 |

Bold: significant correlation without meeting criterion for significant sex effect.

Bold and red: significant correlation and significant effect of sex (p<0.05).

Asterisk (*) designates a significant effect of rearing (p<0.05) within each sex.

**Table 3.** Predictive relationships between PD28 BLA-PFC rsFC and PD48 anxiety-like behavior.

| PD28 rsFC BLA-PL | | PD28 rsFC BLA-IL | |
|---|---|---|---|
| MALE | $R = 0.354$; $R^2 = 0.125$ | MALE | $R = 0.049$; $R^2 = 0.002$ |
| $n = 16$ | p=0.178 | $n = 16$ | p=0.856 |
| FEMALE | $R = 0.428$; $R^2 = 0.175$ | FEMALE | $R = 0.637$; $R^2 = 0.405$ |
| $n = 15$ | p=0.107 | $n = 15$ | p=0.011* |

Bold: significant correlation without meeting criterion for significant sex effect".

current need for more transparency in reporting of ELA paradigms (*Kentner et al., 2019*; *Brenhouse and Bath, 2019*) that pregnant dams were shipped to our facility, therefore all animals in our study underwent shipping stress during gestation. It is thus possible that effects of maternal separation reported here are a consequence of both prenatal and postnatal stress, which is common amongst human instances of early life stress (*Pedersen et al., 2018*). Our findings suggest that females may be particularly vulnerable to neuroanatomical consequences of early adversity, with innervation effects seen earlier in females compared to a later effect observed in males. This is in line with human studies describing sex-dependent effects of early adversity where female participants appear to exhibit more severe adolescent and later-life consequences, especially with regard to affective disorders (*Humphreys et al., 2015*). Rodent studies investigating potential sex differences have also largely revealed female-specific increases in adolescent anxiety-like behavior following ELA (e.g. *Salberg et al., 2019*; *Manzano-Nieves et al., 2018*; *Jin et al., 2018*; *Viola et al., 2019*; but see *Bonapersona et al. (2019b)* for further examination), though some studies have failed to show any anxiety-like effects in adolescence (e.g., *Doherty et al., 2017*). Little, however, has been known to date regarding biological substrates for sex-specific consequences on anxiety-like behaviors.

Our data indicate that BLA-IL innervation increases through adolescence in a bilaminar manner across development, with more dramatic increases following ELA in both males and females. Importantly, these main effects of age support previous work in male rats (*Cunningham et al., 2002*) and now characterize a similar trajectory in female rats. Interestingly, increased innervation to the IL in PD28 females and PD38 males occurred in IL2, whereas previous work has shown that in wild-type adult mice, the BLA preferentially targets amygdala-projecting neurons in IL5 over IL2 (*Cheriyan et al., 2016*). Since IL2 projection neurons do also project back to the BLA (*Gabbott et al., 2005*; *Little and Carter, 2013*), it is possible that atypical hyper-innervation after maternal separation aberrantly targets layer 2, which may drive the altered maturation of BLA-IL functional connectivity we observed in the present work.

While IL innervation showed a developmental increase through adolescence, PL innervation appeared to reach adult-like levels in both sexes earlier than was previously captured (*Van Eden and Uylings, 1985*; *Bouwmeester et al., 2002*). Following ELA, however, our data indicate that aberrant effects on innervation were apparent at PD28 in females, but not until PD38 in males; this finding was observed across both the PL and IL. However, while ELA conferred a similar age-dependent influx of BLA axonal innervation in both regions, the long-term neuroanatomical alterations were different depending on the region examined. In IL there was evidence of precocial maturation, such that ELA induced BLA-IL innervation that was comparable to more mature (PD48) patterns in female juveniles (PD28) and male early adolescents (PD38). Interestingly, in PL we saw a transient effect of ELA that was age- and sex-dependent. Indeed, within the PL we observed that both male and female rats showed an unexpected transient spike in innervation that appeared to resolve at the following developmental timepoint. It is possible that reversal of the innervation seen at earlier time points may be due to pruning mechanisms (*Koss et al., 2014*) that typically occur in adolescence, and these mechanisms may serve to mediate excess innervation (*Rakic et al., 1994*; *Spear, 2000*; *Riccomagno and Kolodkin, 2015*). Prior work has shown a peak in BLA-PL connectivity, specifically at PD30 (*Pattwell et al., 2016*), that may contribute to the present findings of transient ELA-exacerbated hyper-innervation. However, this does not explain why innervation to the PL spiked at PD28, was reduced to CON levels at PD38, and then spiked again by PD48 in female ELA. It is possible that the temporary (one week) isolation following surgery may have acted as a secondary stressor

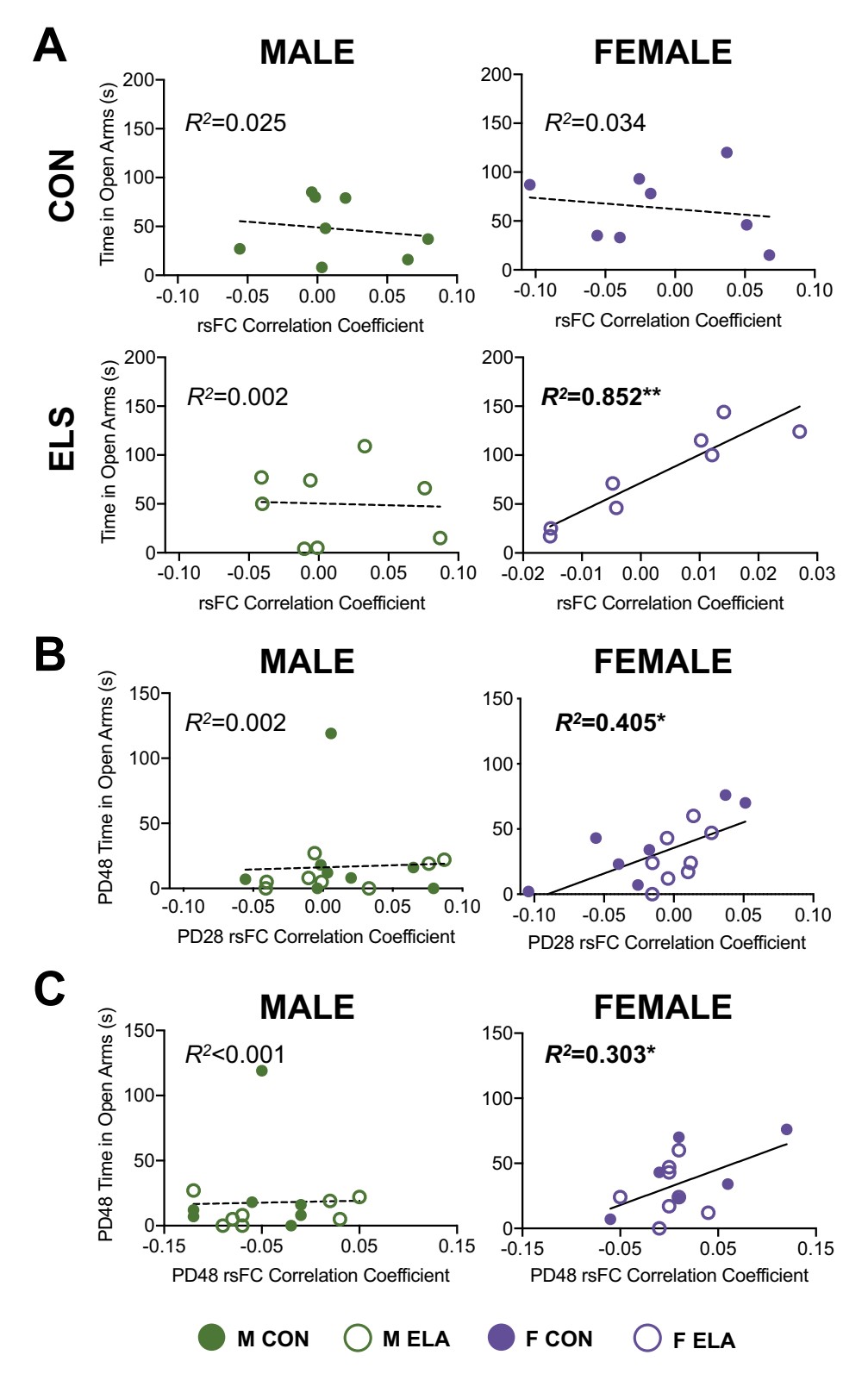

**Figure 6.** ELA significantly impacts the relationship between BLA-IL rsFC and anxiety-like behavior only in females; with PD28 rsFC predictive of PD48 behavior. Results of linear regression analyses to determine the relationship between BLA-IL rsFC correlation coefficients and performance in the EPM (time spent in open arms (seconds) as an index of anxiety-like behavior) are shown for PD28 in (A). Only females with a history of ELA showed a significant correlation, with higher rsFC correlation coefficients correlating with more time spent in the open arms of the EPM. As this data was

*Figure 6 continued on next page*

*Figure 6 continued*

conducted over development in a within-subjects manner, we could further explore predictive relationships between these variables. In line with other rsFC data described here, only female rats showed an overarching predictive effect of early/juvenile (PD28) rsFC correlation coefficients on later (PD48) behavior (B). Indeed, females with a higher rsFC correlation coefficients at PD28 exhibited less anxiety-like behavior (as evidenced by increased time spent in open arms). Furthermore, in the IL females showed a relationship between PD48 rsFC and time spent in open arms at PD48 (C). Individual data points for each animal can be seen on the graphs, with solid circles representing CON cases, and open circles representing ELA cases, with solid regression lines indicative of significant correlations. $n$ = 7-8 per group.*p < 0.05; **p < 0.01".

The online version of this article includes the following source data and figure supplement(s) for figure 6:

**Figure supplement 1.** Similar to findings in *Figure 3* (main article), no effect of rearing on anxiety-like behavior was seen at PD28 or PD48.
**Figure supplement 1—source data 1.** Raw Study 2 Elevated Pluz Maze Data.

acting in conjunction with pubertal changes to produce a resurgence of this ELA-specific phenotype (*Andersen and Teicher, 2008*; *Tzanoulinou and Sandi, 2016*). Relatedly, surgery itself during peri-pubertal development compared to other time periods may have differentially interacted with ELA.

Importantly, the ages at which ELA-exposed females and males first displayed increased BLA-IL innervation (PD28 and PD38, respectively) were the ages at which higher innervation correlated with higher anxiety-like behavior (less time spent in open arms of the EPM). Since monosynaptic input from the BLA to the medial PFC drives aversion to anxiogenic stimuli as measured in the EPM (*Felix-Ortiz et al., 2016*), these data suggest that hypertrophic effects of ELA at the BLA-IL circuit lead to heightened anxiety-like behavior in ELA-exposed animals. However, we were surprised that group-wise comparisons only showed higher anxiety-like behaviors in ELA females at PD38, but not at PD28 when innervation was increased (see *Figure 3*). Group comparisons also failed to reveal higher anxiety-like behaviors in ELA-exposed males at any age. Our group and others have previously observed increased anxiety-like behaviors in male and female rats following ELA (e.g., *Holland et al., 2014*; *Ganguly et al., 2015*; *Jin et al., 2018*), therefore further work will reveal whether more sensitive assays can more reliably demonstrate anxiety-like behavior in ELA-exposed rats. Here we demonstrate that, regardless of mean differences, ELA experience can forge a relationship between heightened innervation and anxiety-like behavior at distinct developmental time-points that could suggest mechanistic ties between early hypertrophy of inputs and behavior.

Transient effects of ELA on innervation and behavior are notable because adolescent perturbations have been shown to have lasting consequences on later life function. Multiple lines of evidence from both clinical and animal studies suggest that physiological or experiential anomalies during critical periods of development can program the central nervous system for susceptibility or resilience to future environments (*Andersen, 2003*; *Nederhof and Schmidt, 2012*). One example is seen in an animal model utilizing prenatal treatment with the mitotoxin MAM, which produces adolescent-specific corticolimbic and mesolimbic dysfunctions that drive a hyperresponsivity to stress with anxiety-like and psychosis-like behavior. If conversion to the dysfunctional phenotype in adolescence is prevented by relieving stress in adolescence, MAM-treated animals will still be more susceptible to affective dysfunction in adulthood (*Gomes et al., 2019*). Moreover, ELA reportedly leads to adolescent alterations in PFC NMDA receptor subunit composition that regulate anxiety-like behaviors in males (*Ganguly et al., 2015*); future work will reveal whether increased glutamatergic input from the BLA drives these receptor changes in the PFC. Since the PFC can serve reciprocally as a modulator of BLA activity (*Rosenkranz and Grace, 2002*), it is therefore possible that transient innervation changes lead to long-term receptor alterations and subsequent changes to the efficiency of BLA-PFC functional connectivity. Indeed, we observed an effect of rearing in female anxiety-like behavior and functional connectivity after the age when innervation to the PL and IL2 was increased (PD28). Together, the early increases in innervation after ELA reported here may perturb normal critical period maturation, leading to late-onset effects on functional connectivity and affective dysregulation.

While the current work did not directly address the causal relationship between innervation and rsFC, results from the rsFC analyses in females supported the idea that early maturation of BLA-IL innervation disrupts later functional relationships between these regions (*Tottenham and Galván, 2016*). Specifically, we report here that females exposed to ELA displayed dampened maturation of BLA-IL rsFC compared to female CON; while no effects of rearing were apparent at PD28, ELA female rats failed to display the increase in BLA-IL rsFC by PD48 that was observed in female CON.

This is in line with previous work showing reduced BLA-mPFC connectivity following a limited bedding model (*Guadagno et al., 2018*). Interestingly, increased [more mature] rsFC BLA-IL connectivity in female ELA at PD28 correlated with decreased anxiety-like behavior at that age, as well as 20 days later. This suggests that more mature connectivity in female juveniles may confer enduring behavioral resilience, which is consistent with previous reports that dampened connectivity is associated with increased anxiety in adolescence (*Kim et al., 2011*; *Nooner et al., 2013*). While ELA exposure in males appeared to result in lower juvenile (PD28) rsFC between the BLA and the PL, no effects on the magnitude of change between PD28-PD48 were noted, therefore altered maturation per se was not observed in males. Interestingly, ELA affected male BLA-PL rsFC at an age that preceded any ELA-attributable increase in innervation, suggesting that the reciprocal connectivity between the two regions was altered without the influence of aberrant amygdalofugal innervation.

ELA experience resulted in dampened maturation of BLA-IL rsFC in females, in contrast to the precocial maturation we observed in axonal innervation of the IL from the BLA, as well as the accelerated maturation of task-based FC reported following childhood maltreatment in humans (*Gee et al., 2013a*). While BLA-mPFC rsFC in humans has been observed to increase towards more positive connectivity through adolescent development (*Gabard-Durnam et al., 2014*), task-based FC during fearful face presentations declines from positive connectivity to a negative connectivity (*Gee et al., 2013b*). This task-based FC was further found to reach mature (more negative) levels in previously institutionalized children (*Gee et al., 2013a*). In contrast, girls who express higher basal cortisol levels at 4.5 years display lower rsFC between the ventromedial PFC and the amygdala at 18 years (*Burghy et al., 2012*), corroborating our current findings in PD48 females (see *Figure 5*). rsFC reflects an intrinsic alternate resonance between different brain areas connected at large scale, in contrast to the more isolated activation of the corresponding brain areas during a task (*Rasero et al., 2018*). Therefore, it appears that early hyperinnervation to the PFC may alter the stability of the functional BLA-PFC network as it develops, leading to increased anxiety when innervation arises and disrupted maturation of rsFC; though how these effects in rats relate to task-based FC is currently unknown. It is also possible that early maturation of BLA-PFC projections may overwhelm the slow-maturing reciprocal projections to the BLA that dampen anxiety-related circuit activity in mature brains (*Arruda-Carvalho et al., 2017*; *Selleck et al., 2018*).

The present findings support previous work indicating the distinct functional relationship between the BLA and PL and IL regions of mPFC (*Calhoon and Tye, 2015*), with PL connectivity related to anxiogenic effects (*Felix-Ortiz et al., 2016*) and IL connectivity promoting anxiolytic effects (*Maroun et al., 2012*). Furthermore, these findings are in line with the idea that altered neuroanatomy and functionality of PL and IL, along with BLA, likely have reciprocal effects on one another, further contributing to the exacerbation of ELA-induced anxiety-like phenotypes in adolescence and early adulthood (*Likhtik and Paz, 2015*; *Zimmermann et al., 2019*).

Taken together, maternal separation in rats was found to disrupt normative development of both anatomical (axonal innervation) and functional (rsFC) connectivity within a circuit regulating emotional processing, with sex-specific effects and evidence of resilience in individuals with precocial maturation of rsFC. Future work will investigate the physiological consequences of increased innervation that underlies rsFC and behavioral effects, as well as the potential impact of puberty on increased innervation, since females also reportedly display accelerated puberty initiation following ELA (*Grassi-Oliveira et al., 2016*). The present findings have implications for intervention based on experience, age, and sex – three functionally interactive factors that uniquely define risk in every individual.

## Materials and methods

### Subjects

For Studies 1 (BLA-mPFC innervation) and 2 (rsFC), timed-pregnant Sprague-Dawley rats (Charles River, Wilmington, MA) arrived at gestational day 15. Rats were housed under standard laboratory conditions in a 12 hr light/dark cycle (lights on at 0700 hr) temperature- and humidity-controlled vivarium with access to food and water ad libitum. Following birth (postnatal day [PD]0), litters were randomly assigned as either: CON, and left undisturbed with the exception of cage-changing twice/week and weighing (PD9, 11, 15, 20); or ELA via maternal and peer separation, as described

previously (*Coley et al., 2019*; *Farrell et al., 2016*; *Ganguly et al., 2019*; *Grassi-Oliveira et al., 2016*; *Wieck et al., 2013*) and below. In preliminary power analyses, we achieved with group sizes of 7–9 subjects effect sizes of η2 = 0.08–0.27 and power of 1-β=0.80–0.98 for piloted anterograde tracing data, and effect sizes of η2 = ~ 0.12 and power of ~1-β=0.80 with group sizes of 22 subjects in behavior studies, therefore we aimed for *n* = 10 for all studies involving innervation, allowing behavioral studies to include animals that did not reach criteria for inclusion in innervation analyses in Study 1. Group sizes were chosen for Study two based on preliminary studies showing that BOLD data from 7 to 8 rats/group yielded Cohen's D values of 1.2–2.5. On PD1 litters were culled to 10 (+ /- 2) pups, maintaining equal ratio of male and female whenever possible, with one rat per litter assigned to each experimental group (i.e. age and sex) to avoid litter effects. Pups were assigned pseudo-randomly to experimental groups, where an investigator not involved in the study assigned a number to each pup in a litter, and a separate investigator assigned one number from each sex to an experimental group. All measurements were conducted by experimenters blinded to experimental condition through coding of microscopic slides, video recordings of behavior, and rsFC files. Throughout all experiments, outliers were identified when subjects were visibly agitated or otherwise behaving atypically outside of experimentation and were excluded from all analyses.

ELA pups were separated from dams and littermates in individual cups with home cage pine shavings in a circulating water bath (37°C) from PD2-PD10. At PD11-20, when body temperature is self-regulated, pups were individually separated into cages. Pups were separated for 4 hr each day (0900 h-1300h) during which time pups were deprived of maternal and littermate tactile stimulation and nursing, but not from maternal odor. ELA dams remained in their home cages but were deprived of their entire litters during separations. Pups were weaned at PD21 into same-sex mixed-litter pairs and left undisturbed until surgery/behavioral assessment for Study 1 (either PD21/28, 31/38, or 41/48; *Figure 7A*). Separate cohorts were used for Study 2 – with treatment identical to Study 1 – and left undisturbed until behavioral assessment and rsFC (PD28, PD48), with subjects imaged at both ages. Experiments were performed in accordance with the 1996 Guide for the Care and Use of Laboratory Animals (NIH) and with approval from Northeastern University's Institutional Animal Care and Use Committee.

## Study 1: BLA-PFC innervation

### Surgeries

Male and female rats from CON and ELA litters at PD21, PD31, or PD41 underwent stereotaxic injections of biotinylated dextran amine (BDA; NeuroTrace 10,000 MW Anterograde Tracer Kit (Invitrogen, Carlsbad, CA); reconstituted to 10% with phosphate buffered saline (PBS)) into the BLA (*Figure 7B*). For maximal uptake without excess bolus, 200 nL (0.2μ l) of BDA was injected into the BLA using a mounted 32-gauge Neuros syringe (Hamilton Company, Reno, NV) attached to a micro-infusion pump (11 Elite Nanomite; Harvard Apparatus, Holliston, MA).

Rats were first anesthetized with Isoflurane in an induction chamber before beginning surgical procedures. Surgical site was shaved, and the animal was secured via ear bars with top incisors positioned over a bite bar within a nose cone to provide continuous Isoflurane anesthetic during surgery. A subcutaneous injection of buprenorphine (0.03 mg/kg body weight) was administered as a postoperative analgesic. Once sufficiently anesthetized, measured via lack of pedal reflex, a skin incision was made along the midline of the skull to visualize Bregma and Lambda. Dorsal-ventral (DV) coordinates for Bregma and Lambda were taken to ensure that the skull was level, and Bregma was used as a landmark to navigate to the position above the BLA, where a small hole was drilled into the right hemisphere of the skull, and the needle was slowly lowered into the BLA (*Figure 7B*). DV depth was calculated from dura surface to account for individual differences in skull thickness. Once needle was lowered to the target, BDA was slowly infused at a rate of 40 nL/min over the course of 5 min and left in place for an additional 5 min to allow for diffusion of BDA solution before being slowly retracted. The incision was sutured, and rats were returned to individual cages and allowed 5–6 days for recovery before behavioral testing, with all brain tissue collected one-week post-surgery.

### Behavioral assessment: Elevated Plus Maze

Elevated Plus Maze (EPM) performance was evaluated 6–7 days post-surgery in all animals. The apparatus was constructed of opaque Plexiglas with four radiating arms (50 cm x 10 cm) around a

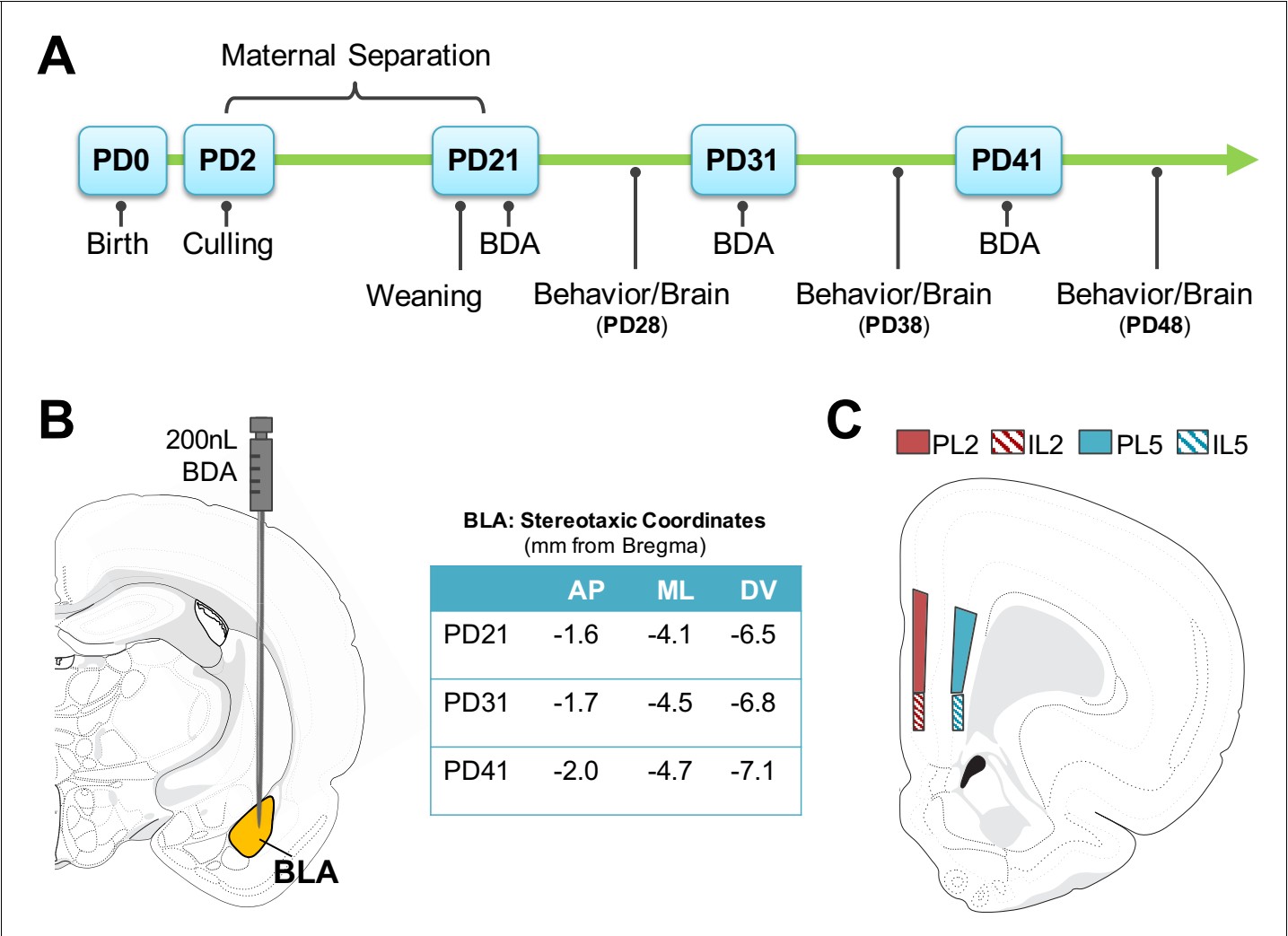

**Figure 7.** Timeline and methodology for study 1. (**A**) Methodological timeline for Study one indicating maternal separation (for ELA groups), weaning timeline, and surgical/behavioral milestones. Biotinylated Dextran Amine (BDA) microinfusions were performed at PD21, PD31, or PD41. Behavior (elevated plus maze; EPM) was performed at PD28, PD38, or PD48, and was followed by brain collection. (**B**) Stereotaxic coordinates for surgeries at each developmental time point and anatomical map of basolateral amygdala (BLA) injection site where 200 nL of BDA was infused via Hamilton Neuros syringe. (**C**) Neuroanatomical map of quantified regions of BLA-PFC axonal innervation. Quantification was conducted via unbiased stereology within the prelimbic (PL) and infralimbic (IL) in layers 2 and 5. Atlas modified from *Swanson (2018)*. AP (anterior-posterior); ML (medial-lateral); DV (dorsal-ventral).

center square (10 cm x 10 cm), with 40cm-high walls surrounding two opposing arms, leaving the other two arms open. Between each animal 50% ethanol solution was used to clean the apparatus. Rats were acclimated to the testing room for 10 min before testing began and were then placed in the center square of the apparatus under red light facing a closed arm. Behavior was recorded for 5 min by an observer blind to condition. Behavioral scoring included: four-paw entries into/time spent (seconds) in open and closed arms, number of arm crossings, and head dips.

## Tissue collection and inclusion criteria

7 days post-surgery, rats were deeply anesthetized with $CO_2$ and transcardially perfused with ice-cold (0.9%) saline, followed by ice-cold 4% paraformaldehyde solution. Brains were extracted and stored in 4% paraformaldehyde solution for 1 week before being transferred to a 30% sucrose solution for 4 days for cryoprotection. All brains were sliced into 40 μm serial sections on a freezing microtome (Leica Biosystems, Wetzlar, Germany), with serial sections placed in well plates filled with

freezing solution for −20°C storage. Sections taken for analysis included: PFC (+5.2 mm through +2.5 mm from Bregma) and BLA (−1.4 mm through −3.6 mm from Bregma) as outlined in *Paxinos and Watson (1997)*. Serial sections were collected such that coronal sections within a single well were separated by approximately 240 μm.

BLA-derived axonal innervation of the PFC, and BLA injection bolus for each animal, was visualized using diaminobenzidine (DAB) as indicated by the manufacturer (NeuroTrace BDA Kit). One well for each animal was used for each brain region (PFC, BLA). Free-floating sections were washed 2 × 10 min in PBS (7.4 pH) on an agitator before being transferred into Avidin solution (1:4000 in 0.3% PBS-T; NeuroTrace BDA Kit) overnight at 4°C on an agitator. Sections were then washed 3 × 5 min in PBS and transferred into 0.05% DAB solution (NeuroTrace BDA kit) for 10 min for processing. Stained sections were washed in PBS, mounted on glass slides and coverslipped with DPX. Adjacent sections were randomly sampled from each experimental group (*n*'s = 4/group) and stained with Cresyl Violet to determine average volume of the BLA based on age, sex, and condition. There were no significant differences based on sex or condition, thus volumes were collapsed across these to determine the average BLA volume per age (*Figure 1—figure supplement 1*).

## Bolus verification in BLA and inclusion criteria
To determine the total volume of the BDA bolus within the BLA at the injection site, six DAB-stained sections containing BLA/BDA bolus were analyzed per animal (inter-slice interval of 240 um). Volumetric analysis was conducted in StereoInvestigator (MBF Bioscience, Wilmington, VT) using a Cavalieri probe to estimate total bolus volume. Different markers were used to identify regions of the bolus that fell within versus outside of the BLA. Approximate percentage of BLA filled with BDA tracer was calculated by dividing the volume of bolus within the BLA by the average total BLA volume, specific to age, for each animal. Only subjects with a bolus filling greater than 60% of the average BLA volume (with no more than 15% of total bolus volume outside of the BLA) were included in analyses. See *Figure 1B,E* for details on bolus size, location, and representative cases.

## Quantification of PFC axonal innervation
Three sections of each brain containing PFC (PL and IL regions: +4.0 mm through +2.8 mm from Bregma; 240 um inter-slice interval) were used for StereoInvestigator analysis. BLA-derived axon fiber length was estimated via hemispheric SpaceBalls Probe in four regions per section: PL layer 2 (PL2) and 5 (PL5), and IL layer 2 (IL2) and 5 (IL5); *Figure 1C*. Axonal fibers in focus over the probe (radius = 10 μm) through the Z plane were identified at 60x oil immersion to determine estimated fiber length (*Gundersen and Jensen, 1987*; *Gundersen et al., 1999*; *Mouton et al., 2002*).

## Statistical analyses of neuroanatomical and behavioral data
To determine effects of sex, rearing, and age on estimated total length of BLA-derived axonal terminals in the PFC, three-way ANOVAs were conducted within each entire region (PL, IL), and each layer (PL2, PL5, IL2, IL5). Layer-specific ANOVAs were analyzed with a Bonferroni alpha adjustment to correct for multiple comparisons, followed by post hoc individual comparisons with corrected p-values reported. Two-way ANOVAs were conducted separately for male and female groups to determine sex-specific effects. EPM performance was also analyzed with 3-way ANOVA and subsequent 2-way ANOVA within each sex. Significant main effects and/or interactions were followed up where appropriate with Bonferroni post-hoc analyses corrected for multiple comparisons. Trending effects (p<0.1 in first-order analyses; p<0.05 in multiple comparisons where alpha adjustments were performed) were followed up with post-hoc comparisons only when moderate-to-strong effect sizes were determined via partial eta squared.

Separate linear regression analyses were conducted in male and female rats to determine whether there were relationships between behavior and BLA-PFC innervation. Fisher's r-to-z transformations were used to assess any significant impact of sex or rearing on the strength of the relationships. Behavioral data collected from all subjects (regardless of whether they later met neuroanatomical inclusion criteria) was included in statistical analyses of EPM. However, all correlational analyses including behavioral data were performed only with the data from cases meeting neuroanatomical inclusion criteria. All statistical analyses were performed in either SPSS v25 (IBM, Armonk, NY) or PRISM 8 (Graphpad Software, San Diego, CA).

## Study 2: Resting state Functional Connectivity (rsFC)

MRI scanning was performed on male and female CON and ELA subjects at PD28 and PD48 in a Bruker BioSpec 7.0T/20 cm USR horizontal magnet (Bruker, Billerica, MA) with a 20 G/cm magnetic field gradient insert (ID = 12 cm) capable of 120μs rise time. Rats were anesthetized and maintained at 1–2% isoflurane and oxygen with a flow rate of 1 L/min throughout scanning, with breathing rate (40–50 breaths per minute) carefully monitored by an investigator and anesthetic levels adjusted accordingly. Rats were scanned at 300 MHz using a quadrature transmit/receive volume coil built into the rat head holder and restraint system (Animal Imaging Research, Holden, MA). rsFC was acquired by gradient-echo triple-shot echo-planar imaging (EPI) pulse sequence with the following parameters: matrix size = $96 \times 96 \times 20$; repetition time (TR)/echo time (TE) = 3000/15msec; voxel size = $0.312 \times 0.312 \times 1.2$ mm; slice thickness = 1 mm; volume = 200. T2-weighted high-resolution anatomical scans were conducted using RARE pulse sequence with imaging parameters as follows: matrix size = $256 \times 256 \times 20$; TR/TE = 4369/12msec; voxel size = $0.117 \times 0.117 \times 1$ mm; slice thickness = 1 mm.

### Pre-Scan Elevated Plus Maze

Prior to both PD28 and PD48 imaging, subjects were evaluated for anxiety-like behavior in the EPM to determine whether it could be predictive of rsFC alterations, and whether juvenile behavior/rsFC might be predictive of later adolescent outcomes in the same rat. Methods identical to Study 1.

Analysis of rsFC and Behavior rsFC was measured using seed-based voxel-wise analysis, with BLA as the seeded region. rsFC between BLA and IL and PL regions of PFC were the focus of the present study. Data analyses were conducted using Analysis of Functional NeuroImages (AFNI_17.1.12; NIH), FMRIB software library (FSL, v5.0.9), and Advanced Normalization Tools (http://stnava.github.io/ANTs/). Resting-state Blood Oxygen-Level Dependent (BOLD) data were used for brain-tissue data extraction using 3Dslicer (https://www.slicer.org). Skull-stripped data were despiked to remove large signal fluctuations due to scanner and physiological artifacts. Slice-timing correction was conducted to correct data from interleaved slice order acquisition. Head motion correction was carried out using six parameters, with first volume as reference slice. Each subject was registered to a standard MRI Rat Brain Template (Ekam Solutions LLC, Boston, MA) using non-linear registration. In order to further reduce motion effects and physiological fluctuations, regressors comprised of six motion parameters, the average BOLD signal in white matter and ventricular regions, as well as motion outliers among all data volumes were fed into a nuisance regression model. Band-pass temporal filtering (0.01 Hz-0.1Hz) and spatial smoothing (FWHM = 0.6 mm) were performed on the residual data followed by signal detrending.

Whole brain voxel-wise Pearson's correlation coefficients were calculated for each subject and were transformed for normality using Fisher's z. Because this experiment was conceived to follow up on observed sex-specific effects on innervation measures, a priori hypotheses drove the use of a two-way mixed ANOVA within each sex (age as a within-subjects factor and stress as a between-subjects factor) in lieu of a three-way ANOVA. Therefore, two-way ANOVAs were conducted for each sex using AFNI 3dLME, with a false discovery rate curve computed. Post-hoc contrast effects focusing rearing condition (CON vs. ELA) at two levels of age (PD28 or PD48), as well as contrasts effects of age at two levels of rearing condition, were conducted where appropriate. Findings with a voxel-wise uncorrected $p<0.005$, with a minimum cluster size of 30 voxels, was regarded as statistically significant.

Linear regressions were conducted to determine the relationship between EPM performance and correlation coefficients of BLA-seeded rsFC data for PL and IL in male and female rats. Since rats were tested at two time points, regressions were also performed to assess whether juvenile outcomes (behavior and/or rsFC) were predictive of later adolescent outcomes. Fisher's r- to- z transformations were performed to determine any significant impact of sex or rearing on the strength of the relationships. All statistical analyses for behavior, as well as regression analyses to determine relationships between rsFC correlation coefficients and behavior, were performed in either SPSS v25 or PRISM 8.

## Acknowledgements

The authors would like to sincerely thank Dr. Jens Foell (Florida State University) and Dr. Michael Rohan (McLean Hospital) for their feedback and suggestions for rsFC figures for this manuscript. The data contained herein has previously been released as a preprint on BioRxiv (DOI: https://www.bio-rxiv.org/content/10.1101/700666v2).

## Additional information

### Competing interests
Craig F Ferris: has a financial interest in Animal Imaging Research, the company that makes the rat imaging system. The other authors declare that no competing interests exist.

### Funding

| Funder | Grant reference number | Author |
| --- | --- | --- |
| National Institute of Mental Health | 1R01MH107556-01 | Heather C Brenhouse |

The funders had no role in study design, data collection and interpretation, or the decision to submit the work for publication.

### Author contributions
Jennifer A Honeycutt, Data curation, Formal analysis, Investigation, Methodology, Project administration; Camila Demaestri, Data curation, Formal analysis, Investigation, Methodology; Shayna Peterzell, Formal analysis, Investigation, Methodology; Marisa M Silveri, Conceptualization, Data curation; Xuezhu Cai, Data curation, Formal analysis; Praveen Kulkarni, Data curation, Formal analysis, Validation, Investigation; Miles G Cunningham, Conceptualization, Funding acquisition, Methodology; Craig F Ferris, Resources, Supervision; Heather C Brenhouse, Conceptualization, Resources, Data curation, Formal analysis, Supervision, Funding acquisition, Methodology, Project administration

### Author ORCIDs
Jennifer A Honeycutt (iD) https://orcid.org/0000-0002-4879-0203
Heather C Brenhouse (iD) https://orcid.org/0000-0001-7591-4964

### Ethics
Animal experimentation: This study was performed in strict accordance with the recommendations in the Guide for the Care and Use of Laboratory Animals of the National Institutes of Health. All of the animals were handled according to approved institutional animal care and use committee (IACUC) protocols (#19-0313R) of Northeastern University. The protocol was approved by the IACUC of Northeastern University (Animal Welfare #: D16-00095). All surgery was performed under isoflurane anesthesia, and every effort was made to minimize suffering.

### Decision letter and Author response
Decision letter https://doi.org/10.7554/eLife.52651.sa1
Author response https://doi.org/10.7554/eLife.52651.sa2

## Additional files

### Supplementary files
- Supplementary file 1. Table of non-significant statistical results illustrated in *Figure 2*.
- Supplementary file 2. Table of non-significant statistical results illustrated in *Figure 3*.
- Transparent reporting form

## Data availability

All data generated for Figures 1, 2, 3, 4, and Figure 6—figure supplement 1 are provided as source data files. Data generated for Figures 5 and 6 will be made available once an ongoing additional analysis is completed for another research report. Data deposited to Dryad, https://doi.org/10.5061/dryad.jdfn2z371.

The following dataset was generated:

| Author(s) | Year | Dataset title | Dataset URL | Database and Identifier |
|---|---|---|---|---|
| Brenhouse HC | 2020 | Data from: Altered corticolimbic connectivity reveals sex-specific adolescent outcomes in a rat model of early life adversity | http://dx.doi.org/10.5061/dryad.jdfn2z371 | Dryad Digital Repository, 10.5061/dryad.jdfn2z371 |

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
