## [Decision Letter]

**Acceptance summary:**

The authors used a combination of neuronal tracing and brain imaging to understand the impact of early-life stress (ELS) on the development of the emotional brain in rats. Consistent with this team's earlier work and with brain imaging studies in humans exposed to ELS, the authors demonstrate that ELS accelerates the maturation of connectivity between the basolateral amygdala and prefrontal cortex, and this was particularly so among females. Imaging measures of functional connectivity were influenced by ELS in females at both ages, but only transiently among males. We were enthusiastic about this paper, in part, because makes an important step toward developing integrative cross-species models of stress-related psychopathology. This kind of work is just what the field needs to build bridges between the human and animal literature. The experiments appear to be technically sound, and the revised manuscript is thorough, clearly written, well-organized, has appropriate statistics, and makes appropriate conclusions.

**Decision letter after peer review:**

Thank you for submitting your article "Too much, too young? Altered corticolimbic connectivity reveals sex-specific outcomes in a rat model of early adversity" for consideration by *eLife*. Your article has been reviewed by three peer reviewers, and the evaluation has been overseen by Alexander Shackman as the Reviewing Editor and Kate Wassum as the Senior Editor. The following individual involved in review of your submission has agreed to reveal their identity: Cate Pena (Reviewer #2).

The reviewers have discussed the reviews with one another and the Reviewing Editor has drafted this decision to help you prepare a revised submission.

Summary:

The authors used a combination of anterograde axonal tracing and resting state functional connectivity to determine whether early life stress (ELS; maternal deprivation) altered trajectories of BLA-medial PFC (PL/IL) circuit development in male and female rats. Consistent with the lab's earlier work examining PFC interneuron maturation, and with regional connectivity findings from ELS-exposed humans, the authors find ELS accelerates maturation of BLA-PFC in females at an earlier time point than males. ELS also accelerated BLA-PFC maturation among males, albeit at a later age. Resting state connectivity was altered by ELS in females at both ages, but only transiently among males.

The reviewers and I were enthusiastic about the manuscript, which makes an important step toward developing integrative cross-species models of stress-related psychopathology:

• It is a nice series of experiments that take on the challenge of actually testing a long held belief about early life stress, i.e. accelerates development.

• What I really like about this manuscript is the use of a human focused technique that has been one basis for the idea of stress-induced precocial brain development (rsFMRI) and the authors paired this with experiments showing accelerated development in some more specific neural measures indicative of developmental maturity.

• Another strength of this manuscript is the use of fMRI paired with a more refined technique – this is just what the field needs to build bridges between the human and animal literature.

• This work is an important extension of several earlier findings across species using novel approaches, and includes female rats (absent in much previous work).

• Overall, the manuscript is thorough, well-organized, has appropriate statistics, and makes appropriate conclusions.

• I reviewed an earlier version of this manuscript elsewhere. The figures and clarity of the current version are much improved. The authors addressed my concerns in the Discussion and should be commended for prudence with language and interpretation.

• The experiments are technically sound, the rationale for the study is clear, and the results are appropriately analyzed.

• These studies were performed with care and expertise.

Major concerns and suggestions:

Nevertheless, our enthusiasm was somewhat tempered by several key limitations of the report. Major concerns included: weakness of the behavioral data, and important inconsistencies between the behavioral and neural data.

Weakness:

• Figure 3. I do not see an anxiety effect for ELS – there is one data point that is significant taken from 3 measures in the EM over 3 ages – the significant point is at PD38 female ELS is lower than controls for time spent in open arm. Looking at the data, it seems this is due to one outlier in the control group with a very high score – all other data points are in the same general range. I understand the statistics presented in the manuscript show significant effects but, something seems off.

• Barring stronger behavioral data, the correlations between defensive behaviors (behavior) and brain measures-a major focus of the manuscript-seem unjustified.

• It seems like there is a small behavioral effect at one age but only one sex at that age. Visual inspection (female, control on PD38) raises the possibility the effect would disappear if one control data point was removed – leaving no effect at any age for any sex. Without the behavioral effect, most of the analysis in the manuscript needs to be deleted (i.e. correlations between neural measures and behavior).

Inconsistencies Across Measures:

• This (BLA-PL) came across as the strongest hypothesis from the start. ELS would be expected to increase anxiety over the lifetime and alteration of the BLA-PL would be the most straightforward mechanism to achieve this. By far, the ELS on BLA innervation of the PL were the clearest finding. ELS females showed earlier innervation (on P28) that receded on P38 but reemerged on P48. Males showed the exact inverse pattern, with ELS males showing greater innervation only on P38. These results set up a clear prediction for behavior in the elevated plus maze. Females should show greater anxiety on P28 and P48, but not P38; males should show the inverse pattern. This did not happen, there was no effect of ELS on male EPM behavior. In females, there was a main effect of ELS, but oddly, this was greatest on P38 – the developmental window during which BLA-PL connectivity did not differ. To me, this indicates that effect of ELS on innervation and behavior are not prominent. Further, when the relationship between connectivity and behavior were examined, a correlation and sex difference only emerge on P38. This was driven by a lack of relationship in females (which showed an effect of early life stress on that day) but a positive relationship in males (which showed no effect of early life stress on that day).

• fMRI study. Again, there was no relationship between BLA-PL connectivity and behavior in the EPM. Thus, the pathway most clearly affected by ELS in a sex-specific manner does not appear to contribute to anxiety behavior. Relationships were found for BLA-IL innervation and behavior, however the case for a sex-specific effect of ELS on BLA-IL innervation was much weaker. No three-way interaction was observed, making it questionable to split up males and females for separate analyses. Any likely BLA-IL innervation effects were layer-specific and observed in males, yet control and ELS males did not differ in the elevated plus maze.

• The effects of ELS on BLA-PFC innervation/connectivity seem to have little or nothing to do with the effects of ELS on anxiety behavior. On one level this is not surprising, ELS must be affecting many brain regions and networks. However, these results seem to indicate that BLA to PFC projections are not critical pathways within this network.

• For females, the authors only report a main effect of ELS. ELS females spent less time in the open arms on all developmental days. Changes in BLA-PFC connectivity do not appear to drive this effect because they show a completely different developmental pattern. If anything, innervation effects are weakest on P38, yet anxiety behavior is most different on P38 (although this could change depending on how the authors address the outlying data point noted above).

Given these concerns, we have some specific suggestions for enhancing the report:

• The authors need to address whether the one positive behavioral mean difference reflects a single case. Barring stronger behavioral data, the correlations between defensive behaviors (behavior) and brain measures-a major focus of the manuscript-seem unjustified and should be omitted.

• Focus on the neuroscience: The neuroscience adds a level of specificity to the existing literature, but it needs to be integrated with the other animal literature – we know a lot about PFC-BLA anatomical development and its disruption by early life stress. There is a good summary of the human literature in the paper already. Can't we use this new animal data to help us understand what the developmental changes in PFC-BLA might mean?

• A strength of this manuscript is the use of fMRI paired with a more refined technique – this is just what the field needs to build bridges between the human and animal literature. The manuscript just needs to go the next step of integrating the results with the animal literature to better characterize what this means for neural processing of fear and other amygdala-dependent behaviors.

---

## [Author Response]

Major concerns and suggestions.Nevertheless, our enthusiasm was somewhat tempered by several key limitations of the report. Major concerns included: weakness of the behavioral data, and important inconsistencies between the behavioral and neural data.Weakness:• Figure 3. I do not see an anxiety effect for ELS – there is one data point that is significant taken from 3 measures in the EM over 3 ages – the significant point is at PD38 female ELS is lower than controls for time spent in open arm. Looking at the data, it seems this is due to one outlier in the control group with a very high score – all other data points are in the same general range. I understand the statistics presented in the manuscript show significant effects but, something seems off.• It seems like there is a small behavioral effect at one age but only one sex at that age. Visual inspection (female, control on PD38) raises the possibility the effect would disappear if one control data point was removed – leaving no effect at any age for any sex.

We agree with the reviewers that the major strength of this article is the juxtaposition of altered anatomical innervation with a translational measure of altered functional connectivity in rats exposed to early life adversity, and we believe that a behavioral correlate provides useful additional information about the translatability and functional significance of these findings. With regards to the concern that one data point was driving the effect of maternal separation on female P38 time in open arms, we re-ran our 3-way analysis with that point removed and the effects held up (significant sex x rearing interaction *p* = 0.039; main effect of rearing *p* = 0.006; 2-way ANOVA main effect of rearing in females *p* <0.0001). It is possible that the cluster of ELS data points on top of each other toward the low end of the scale make it difficult to perceive the real group difference. We did not remove that one control data point pointed out by the reviewers from our presented analyses, since it did not meet criterion as an outlier using the grubbs test. However, we still agree that our behavioral data, due to a current dearth of highly robust measures of anxiety-like behavior in rodents, has less value than the innervation and connectivity measures, which are robust. Available measures often only pick up “extreme” instances of anxiety when this – often subtle – affective state can be overtly measured in behavioral output (i.e. Steimer, 2011), and the output (particularly in models of chronic stress) can be vastly variable (D'Aquila et al., 1994; Lezak et al., 2017). However, we do believe the behavioral data is still worthy of inclusion, and of correlation with our neural measures. Specifically, there is value in the ability to view our data in the context of other animal studies of anxiety (using the EPM) following maternal separation, given the fact that this is the first study to our knowledge measuring this behavior over development in both males and females. As part of other suggestions raised, our revised manuscript now includes a more in-depth discussion (Discussion section) of the evidence driving the reasoning behind measuring anxiety-like behavior as a correlate of BLA→PFC innervation, regardless of whether or not the data perfectly map to what would be expected. Finally, we believe it is important to show these data in order to most transparently display the true effect size (in the context of all studies) of maternal separation on EPM behavior.

We apologize for the omission of EPM analyses in the functional connectivity cohort and thank the reviewer for pointing this out. Our intent was simply to not be redundant, especially since the cohort used for functional connectivity was a good deal smaller than that used for the first experiment. However, we understand that it is important to analyze these data similarly, and now include a mixed 3-way ANOVA for EPM in these animals. It is important to note that functional connectivity animals were only scanned and run on the EPM at P28 and again (in a longitudinal design) at P48—not at P38—and again no maternal separation effects were found at those ages. As described below, we believe that showing correlations of functional connectivity with this behavior, even at ages when no group differences were found, is informative for illustrating relationships between inter-individual variability in two potentially related measures.

• Barring stronger behavioral data, the correlations between defensive behaviors (behavior) and brain measures-a major focus of the manuscript-seem unjustified.Without the behavioral effect, most of the analysis in the manuscript needs to be deleted (i.e. correlations between neural measures and behavior).

We submit that correlation of behavioral data with innervation measures is important regardless of strong group differences, since we did not necessarily think that there would be vast differences in anxiety-like behavior, and in fact, previous work from our lab and others often suggest small effects of ELS on behavior, which are usually quite variable. Therefore, we sought to correlate behavior with neuroanatomical and function outcomes to determine the gradient of behavioral impairment (measured via anxiety-like behavior) alongside a physiological gradient of ELS-impairment. Here, our correlation/regression analyses show compelling evidence that – despite not always showing significant group-based changes in overt behaviors – we can actually show a relationship (and in some instances – such as change in rsFC across development – even show predictive value of rsFC on behavioral output). We believe that the inclusion of these correlations – particularly where there are no overt effects in behavior due to individual variability – allow for the translational investigation of individual patterns of both neural and behavioral measures in a manner that is comparable to that seen in the human literature.

Inconsistencies Across Measures:• This (BLA-PL) came across as the strongest hypothesis from the start. ELS would be expected to increase anxiety over the lifetime and alteration of the BLA-PL would be the most straightforward mechanism to achieve this. By far, the ELS on BLA innervation of the PL were the clearest finding. ELS females showed earlier innervation (on P28) that receded on P38 but reemerged on P48. Males showed the exact inverse pattern, with ELS males showing greater innervation only on P38. These results set up a clear prediction for behavior in the elevated plus maze. Females should show greater anxiety on P28 and P48, but not P38; males should show the inverse pattern. This did not happen, there was no effect of ELS on male EPM behavior. In females, there was a main effect of ELS, but oddly, this was greatest on P38 – the developmental window during which BLA-PL connectivity did not differ. To me, this indicates that effect of ELS on innervation and behavior are not prominent. Further, when the relationship between connectivity and behavior were examined, a correlation and sex difference only emerge on P38. This was driven by a lack of relationship in females (which showed an effect of early life stress on that day) but a positive relationship in males (which showed no effect of early life stress on that day).• fMRI study. Again, there was no relationship between BLA-PL connectivity and behavior in the EPM. Thus, the pathway most clearly affected by ELS in a sex-specific manner does not appear to contribute to anxiety behavior. Relationships were found for BLA-IL innervation and behavior, however the case for a sex-specific effect of ELS on BLA-IL innervation was much weaker. No three-way interaction was observed, making it questionable to split up males and females for separate analyses. Any likely BLA-IL innervation effects were layer-specific and observed in males, yet control and ELS males did not differ in the elevated plus maze.• The effects of ELS on BLA-PFC innervation/connectivity seem to have little or nothing to do with the effects of ELS on anxiety behavior. On one level this is not surprising, ELS must be affecting many brain regions and networks. However, these results seem to indicate that BLA to PFC projections are not critical pathways within this network.• For females, the authors only report a main effect of ELS. ELS females spent less time in the open arms on all developmental days. Changes in BLA-PFC connectivity do not appear to drive this effect because they show a completely different developmental pattern. If anything, innervation effects are weakest on P38, yet anxiety behavior is most different on P38 (although this could change depending on how the authors address the outlying data point noted above).

As explained above, the behavioral data on P38 does indeed hold up even without the visual outlier, so the effect of maternal separation at that age is real. That said, we appreciate the complexity that derives from correlations between innervation and behavior at PD28 in females and PD48 in males. As the reviewers point out, the correlations in females are consistent with the age at which innervation is most different in maternally separated females, but not at the age when anxiety-like behavior is most strongly affected by rearing (PD38). We agree that these findings would be easier to interpret if both innervation and anxiety were affected by maternal separation at the same age and were correlated at that same age. However, our data in females appear to suggest that, 1) anxiety-like behavior that has been previously found to be regulated by BLA-PFC communication (now discussed at greater lengths in the Introduction) appears later than heightened innervation, which could be interpreted as a consequence of early innervation; and 2) prior to group-wide effects on anxiety-like behavior, increased BLA-IL innervation in maternally separated juveniles correlates with increased anxiety-like behavior in individual subjects. In males, effects on both innervation and anxiety were weaker than in females (which is consistent with data in humans), yet again, a correlation between anxiety-like behavior and BLA→IL innervation was seen at the same age when innervation was heightened in males. The revised manuscript includes a brief discussion more specifically highlighting these interpretations, which we believe adds to the transparency and strength of the revised manuscript.

We also appreciate the observations offered by the reviewer that effects of maternal separation on BLA→IL innervation was seen more clearly at the layer-specific level. We observed that females had hyper-innervation of IL2 at PD28, while males had hyper-innervation of IL2 at PD38. The revised manuscript now includes a brief discussion about the findings that in wild-type mice, projections from the BLA to IL neurons that reciprocally project back to the BLA preferably terminate in L5 over L2 (Cheriyan et al., 2016), while both L2 and L5 IL neurons do both project to the BLA (Little and Carter, 2013; Gabbott et al., 2005). This suggests that atypical hyper-innervation after maternal separation aberrantly targets layer 2, which could have implications for emotional regulation.

Similar to correlations with innervation in the IL, there were significant correlations in females between anxiety-like behavior and resting-state functional connectivity in the IL, where maternal separation had an effect on both BOLD correlations at PD48 and on the PD28-PD48 change in functional connectivity in females. Together, the BLA-IL circuit does appear to be significantly impacted by maternal separation and, given the reciprocal connections between IL2 and the BLA, it is likely that hyperinnervation to IL2 drives aberrant maturation of BLA-IL functional connectivity. This is described more clearly in the revised discussion.

Given these concerns, we have some specific suggestions for enhancing the report:• The authors need to address whether the one positive behavioral mean difference reflects a single case. Barring stronger behavioral data, the correlations between defensive behaviors (behavior) and brain measures-a major focus of the manuscript-seem unjustified and should be omitted.• Focus on the neuroscience: The neuroscience adds a level of specificity to the existing literature, but it needs to be integrated with the other animal literature – we know a lot about PFC-BLA anatomical development and its disruption by early life stress. There is a good summary of the human literature in the paper already. Can't we use this new animal data to help us understand what the developmental changes in PFC-BLA might mean?• A strength of this manuscript is the use of fMRI paired with a more refined technique – this is just what the field needs to build bridges between the human and animal literature. The manuscript just needs to go the next step of integrating the results with the animal literature to better characterize what this means for neural processing of fear and other amygdala-dependent behaviors.

Thank you for appreciating the value of the current work. While many studies using ELS report effects in both the prefrontal cortex and amygdala, very few have directly investigated their communication or connectivity following ELS in rodents, and there have been no work looking at how this might be impacted over development in males and females. The revised Introduction now more thoroughly highlights the animal literature that served as the scientific premise for the current study (now using the terminology early life adversity [ELA] instead of ELS, as per reviewer comments), as well as the fundamental significance of reverse translation, with use of the suggested citations. In the Discussion, we now also refer to the specific strength of this study as pairing fMRI with a more refined technique, which we believe highlights the impact of our work.